# Online Algorithm for Unsupervised Sequential Selection with Contextual Information

**Arun Verma**
Department of IEOR
IIT Bombay, India
v.arun@iitb.ac.in

**Manjesh K. Hanawal**
Department of IEOR
IIT Bombay, India
mhanawal@iitb.ac.in

**Csaba Szepesvári**
DeepMind/University of Alberta
Alberta, Canada
szepi@google.com

**Venkatesh Saligrama**
Departmetn of ECE
Boston University, USA
srv@bu.edu

## Abstract

In this paper, we study *Contextual Unsupervised Sequential Selection* (USS), a new variant of the stochastic contextual bandits problem where the loss of an arm cannot be inferred from the observed feedback. In our setup, arms are associated with fixed costs and are ordered, forming a cascade. In each round, a context is presented, and the learner selects the arms sequentially till some depth. The total cost incurred by stopping at an arm is the sum of fixed costs of arms selected and the stochastic loss associated with the arm. The learner's goal is to learn a decision rule that maps contexts to arms with the goal of minimizing the total expected loss. The problem is challenging as we are faced with an unsupervised setting as the total loss cannot be estimated. Clearly, learning is feasible only if the optimal arm can be inferred (explicitly or implicitly) from the problem structure. We observe that learning is still possible when the problem instance satisfies the so-called 'Contextual Weak Dominance' (CWD) property. Under CWD, we propose an algorithm for the contextual USS problem and demonstrate that it has sub-linear regret. Experiments on synthetic and real datasets validate our algorithm.

## 1  Introduction

Industrial systems, such as those found in medical, airport security, and manufacturing, utilize a suite of tests or classifiers for monitoring patients, people, and products. Tests have costs with the more intrusive and informative ones resulting in higher monetary costs and higher latency. For this reason, they are often organized as a classifier cascade (Chen et al., 2012; Trapeznikov and Saligrama, 2013; Wang et al., 2015), so that new input is first probed by an inexpensive test then a more expensive one. The goal of a cascaded system is to resolve easy to handle examples early so that the overall system maintains high accuracy at low average costs.

Over time, due to environmental changes or test calibrations, sequential testing protocols (STP) may no longer be accurate, resulting in higher costs. While one can leverage off-line methods such as supervised training of cascades (Wang et al., 2015), they require new annotated data collection. In many scenarios, new data cannot be collected in-situ, and system shutdown is not an option. In the absence of annotated data, we face a dilemma. While we can observe test outcomes, we cannot ascertain their reliability due to the absence of ground truth, necessitating *unsupervised sequential selection (USS)* methods, where an arm represents a test/classifier. Recent works (Hanawal et al., 2017; Verma et al., 2019a, 2020a) propose methods for solving the USS problem; however, they

focus exclusively on the non-contextual setting, which in essence requires inputs (people, objects, or products) to be homogeneous, and as such, these methods are unrealistic since contexts (high vs. low risk) can guide the arm selection.

In this context, we propose the contextual USS. In our setup, inputs arrive sequentially, and the learner observes a continuous-valued context as input. While the learner knows the costs of each arm, he does not know the associated stochastic loss. Furthermore, the learner does not benefit from feedback from his arm selection, in contrast to the conventional contextual bandit works (Beygelzimer et al., 2011). Thus, while being agnostic to the true loss, the learner must sequentially choose the arm that leads to the smallest total loss, where the total loss is the sum of the cost of using an arm and the mean loss associated with the arm. As such, our proposed problem is a special case of the stochastic partial monitoring problem with contextual inputs (Lattimore and Szepesvári, 2020, Chapter 37). Most of the prior work on partial monitoring problem is restricted to cases where observed feedback can identify the losses for selected actions. However, in many areas like crowd-sourcing (Bonald and Combes, 2017; Kleindessner and Awasthi, 2018), resource allocation (Verma et al., 2019b), medical diagnosis (Verma et al., 2020b), and many others, feedback from actions may not even be sufficient to identify the losses.

While we draw upon several concepts introduced in earlier work (Hanawal et al., 2017), there are additional challenges in the contextual case due to the unsupervised nature of the problem. First, unlike vanilla-USS, the loss here is context-dependent. We propose notions of contextual weak dominance as a means to relate observed disagreements to differences in losses between any two arms. We then propose a parameterized Generalized Linear Model (GLM) to model the context-conditional disagreement probability between any two arms and validate the model empirically.

A fundamental technical challenge is in the estimation of disagreement probabilities uniformly across all contexts in the finite time while ensuring sufficient exploration between different arm selection protocols, required for honing in on the optimal selection strategy. In particular, since contexts are continuous-valued, and because we have no control over inputs, the contextual observations, in the finite time, may not persistently span the whole space, and estimates are often unreliable. To this end, we adapt techniques from parameterized contextual bandits (Chu et al., 2011; Li et al., 2017) for our unsupervised setting. We propose an algorithm based on the principle of optimism, namely, the larger indexed arm in cascade is chosen when uncertain. We show that our algorithm navigates the exploration-exploitation tradeoffs in different ways and lead to sub-linear cumulative regret. We then validate it on several problem instances derived from synthetic and real datasets.

**Related Work.** *Stochastic Contextual multi-armed Bandits (SCB):* In each round, the learner observes the context and decides which arm, among a finite number of arms, to apply (Beygelzimer et al., 2011). By playing an arm, the learner observes a stochastic reward that depends on the context and the arm selected. The most commonly studied model assumes that each arm is parameterized, and the mean reward of an arm is the inner product of the context and an unknown parameter associated with the arm. Contextual bandits have been applied to problems ranging from online advertising (Li et al., 2010; Chu et al., 2011) and recommendations (Langford and Zhang, 2008) to clinical trials (Woodroofe, 1979) and mobile health (Tewari and Murphy, 2017). *Generalized linear models (GLM)* assume that the mean reward is a non-linear link function of the inner product between the context vector and the unknown parameter vector (Filippi et al., 2010; Li et al., 2017). GLMs are also useful models for the classification problems where rewards, in the context of online learning problems, could be binary (Zhang et al., 2016; Jun et al., 2017). A more challenging non-parameterized version of the stochastic contextual bandits is studied in (Agarwal et al., 2014).

Another framework that is closely related to SCB is *stochastic linear bandits (SLB)* (Auer, 2002; Dani et al., 2008; Rusmevichientong and Tsitsiklis, 2010; Abbasi-Yadkori et al., 2011). In this setup, the environment is parameterized, and there could be uncountably many arms (within some bounded radius), also referred to as decision set. The arms are characterized into their feature vectors, and the mean reward for playing an arm is given as the inner product of the parameter (unknown) and the feature vector associated with the arm. In situations where the decision set is allowed to vary in each round and are finite, SLBs are equivalent to SCBs, where feature vectors correspond to context-arm pairs (Li et al., 2010, 2017). For our work, we leverage GLMs as models for disagreement probability between any two arms. While it is tempting to reduce contextual USS to SCBs, note that, unlike prior works, we do not observe loss for our action choices, and so conventional algorithms such as LinUCB and UCB-GLM (Li et al., 2010; Agarwal et al., 2014; Li et al., 2017) cannot be applied.

Most of the prior work (Hanawal et al., 2017; Verma et al., 2019a, 2020a) considered the problem of learning an optimal action but ignored the contextual information. In this work, we incorporated contextual information, which is readily available in many applications. Exploiting the *real-valued* contextual information (features) for improving the arm selection strategy is non-trivial due to the unsupervised nature of the problem where the standard analysis of contextual bandits does not apply. We made necessary modeling assumptions to leverage GLMs to parameterize the disagreement probability between two arms and extended the existing definitions to address the new setup's learnability issues. However, the problem still requires new ideas and analysis methods to derive an efficient algorithm, which poses new technical challenges for analysis.

## 2 Problem Setting

We consider a stochastic contextual bandits problem with $K$ arms. The set of arms is denoted as $[K]$ where $[K] \doteq \{1, 2, \ldots, K\}$. In each round $t$, the environment generates a vector $\left(X_t, Y_t, \{Y_t^i\}_{i \in [K]}\right)$. The vector $X_t$ denotes the context in round $t$ and forms an independent and identically distributed (IID) sequence drawn from a bounded set $\mathcal{X} \subset \mathbb{R}^d$ according to an unknown but fixed distribution $\nu$. The binary reward for context $X_t$ is denoted by $Y_t \in \{0, 1\}$, which is hidden from the learner. The vector $\left(\{Y_t^i\}_{i \in [K]}\right) \in \{0, 1\}^K$ represents observed feedback at time $t$, where $Y_t^i$ denotes the feedback observed after playing arm $i$ with $X_t$ as input[1]. We denote the cost for using arm $i$ as $c_i \geq 0$ that is known and the same for all contexts.

In contextual USS, the arms are assumed to be ordered and form a cascade. When the learner selects an arm $i \in [K]$, the feedback from all arms till arm $i$ in the cascade are observed. The expected loss of playing the arm $i$ for a given context $x_t$ is denoted as $\gamma_i(x_t) \doteq \mathbb{E}\left[\mathbb{1}_{\{Y_t^i \neq Y_t | X = x_t\}}\right] = \mathbb{P}\left\{Y_t^i \neq Y_t | X = x_t\right\}$, where $\mathbb{1}_{\{A\}}$ denotes indicator of event $A$. For soundness, we assume that the probability density function of context distribution is strictly positive on $\mathcal{X}$ such that the conditional probabilities are well defined. The total expected loss incurred by playing arm $i$ for context $x_t$ is defined as $\gamma_i(x_t) + \lambda_i C_i$, where $C_i \doteq c_1 + \ldots + c_i$ and $\lambda_i$ is a trade-off parameter that normalizes the incurred cost and the loss of playing arm $i$.

Since the true rewards are hidden from the learner, the expected loss of an arm cannot be inferred from the observed feedback. We thus have a version of the stochastic partial monitoring problem (Cesa-Bianchi et al. (2006); Bartók and Szepesvári (2012); Bartók et al. (2014), and we refer to it as contextual unsupervised sequential selection (USS). Let $\boldsymbol{Q}$ be the unknown joint distribution of $(X, Y, Y^1, Y^2 \ldots, Y^K)$. Henceforth we identify a contextual USS instance as $P \doteq (\boldsymbol{Q}, \boldsymbol{c})$ where $\boldsymbol{c} \doteq (c_1, c_2, \ldots, c_K)$ is the known cost vector of arms. We denote the collection of contextual USS instances as $\mathcal{P}_{\text{USS}}$. For instance $P \in \mathcal{P}_{\text{USS}}$, the optimal arm for a context $x_t$ is given as follows:

$$i_t^\star \in \max \left\{ \arg\min_{i \in [K]} \left(\gamma_i(x_t) + \lambda_i C_i\right) \right\}, \tag{1}$$

where the choice of $i_t^\star$ is risk-averse as we prefer the arm with lower error among the optimal arms.

The interaction between the environment and a learner is given in Algorithm 1.

---

**Algorithm 1** Learning on contextual USS instance $(\boldsymbol{Q}, \boldsymbol{c})$

---
For each round $t$:

1. **Environment** chooses a vector $(X_t, Y_t, \{Y_t^i\}_{i \in [K]}) \sim \boldsymbol{Q}$.

2. **Learner** observes a context $X_t = x_t$ and selects an arm $I_t \in [K]$ to stop in cascade.

3. **Feedback and Loss:** The learner observes feedback $(Y_t^1, Y_t^2, \ldots, Y_t^{I_t})$ and incurs a total loss $\mathbb{1}_{\{Y_t^i \neq Y_t | X = x_t\}} + \lambda_{I_t} C_{I_t}$.

The learner's goal is to find an arm for each context such that the cumulative expected loss is minimized. Specifically, for $T$ contexts, we measure the performance of a policy that selects an arm $I_t$ for a context $x_t$ in terms of regret given by

$$\mathfrak{R}_T = \sum_{t=1}^{T} \left( \gamma_{I_t}(x_t) + \lambda_{I_t} C_{I_t} - \left( \gamma_{i_t^\star}(x_t) + \lambda_{i_t^\star} C_{i_t^\star} \right) \right). \tag{2}$$

We seek policies that yield sub-linear regret, i.e., $\mathfrak{R}_T/T \to 0$ as $T \to \infty$. It implies that the learner collects almost as much reward in the long run as an oracle collects that knew the optimal arm for every context. We say that a problem instance $P \in \mathcal{P}_{\text{USS}}$ is learnable if there exists a policy such that $\lim_{T \to \infty} \mathfrak{R}_T/T = 0$.

In the sequel, we discuss the selection criteria for optimal arm for a given context and the conditions under which instances of $\mathcal{P}_{\text{USS}}$ are learnable.

## 2.1 Contextual Weak Dominance

Next, we introduce the contextual weak dominance property of a problem instance.

**Definition 1** (Contextual Weak Dominance (CWD)). *Let $i_t^\star$ denote optimal arm for context $x_t$. Then the context $x_t$ is said to satisfy weak dominance* (WD) *property if*

$$\forall j > i_t^\star : C_j - C_{i_t^\star} > \mathbb{P}\left\{ Y_t^{i_t^\star} \neq Y_t^j | X = x_t \right\}. \tag{3}$$

*A problem instance $P \in \mathcal{P}_{\text{USS}}$ is said to satisfy the* CWD *property if all contexts of $P$ satisfy* WD *property. We denote the set of all instances in $\mathcal{P}_{\text{USS}}$ that satisfies* CWD *property by $\mathcal{P}_{\text{CWD}}$.*

In the following, we use an alternative characterization of the CWD property, given as

$$\xi(x_t) \doteq \min_{j > i_t^\star} \left\{ C_j - C_{i_t^\star} - \mathbb{P}\left\{ Y_t^{i_t^\star} \neq Y_t^j | X = x_t \right\} \right\} > 0. \tag{4}$$

We define $\xi \doteq \inf_{x \in \mathcal{X}} \xi(x)$ and assume that $\xi > 0$. The larger the value of $\xi$, 'stronger' is the CWD property, and easier it is to identify an optimal arm for given contexts. We later characterize the regret upper bounds of proposed algorithms in terms of $\xi$. We also discuss the case when a fraction of contexts satisfies WD property in the supplementary material.

## 2.2 Selection Criteria for Optimal Arm

Without loss of generality, we set $\lambda_i = 1$ for all $i \in [K]$ as their value can be absorbed into the costs. Since $i_t^\star = \max\left\{ \arg\min_{i \in [K]} (\gamma_i(x_t) + C_i) \right\}$, it must satisfy following equation:

$$\forall j < i_t^\star : C_{i_t^\star} - C_j \leq \gamma_j(x_t) - \gamma_{i_t^\star}(x_t), \tag{5a}$$

$$\forall j > i_t^\star : C_j - C_{i_t^\star} > \gamma_{i_t^\star}(x_t) - \gamma_j(x_t). \tag{5b}$$

As the loss of an arm is not observed, the above equations can not lead to a sound arm selection criteria. We thus have to relate the unobservable quantities in terms of the quantities that can be observed. In our setup, we can compare the feedback of two arms, which can be used to estimate the disagreement probabilities between them. For notation convenience, we define $p_{ij}^{(t)} \doteq \mathbb{P}\left\{ Y_t^i \neq Y_t^j | X = x_t \right\}$ for $i < j$. The value of $p_{ij}^{(t)}$ can be estimated as it is observable. Our next result bounds unobserved error rates differences in terms of their observable disagreement probabilities for a given context.

**Lemma 1.** *For any $i$, $j$, and $x_t \in \mathcal{X}$, $\gamma_i(x_t) - \gamma_j(x_t) = p_{ij}^{(t)} - 2\mathbb{P}\left\{ Y_t^i = Y_t, Y_t^j \neq Y_t | X = x_t \right\}$.*

The detailed proof of Lemma 1 and all other missing proofs appear in the supplementary material.

Now, using Lemma 1, we can replace Eq. (5a) by

$$\forall j < i_t^\star : C_{i_t^\star} - C_j \leq p_{ji_t^\star}^{(t)}, \tag{6}$$

which only has observable quantities. For $j > i_t^\star$, using the CWD property, we replace Eq. (5b) by

$$\forall j > i_t^\star \, : \, C_j - C_{i_t^\star} > p_{i_t^\star j}^{(t)}. \tag{7}$$

Using Eq. (6) and Eq. (7), our next result gives the optimal arm for a given context $x_t$.

**Lemma 2.** *Let $P \in \mathcal{P}_{CWD}$ and $\mathcal{B}_t = \left\{ i : \forall j > i, C_j - C_i > p_{ij}^{(t)} \right\} \cup \{K\}$. Then the arm $I_t = \min(\mathcal{B}_t)$ is the optimal arm for a context $x_t$.*

By construction, the optimal arm lies in set $\mathcal{B}_t$. Because of Eq. (6), any sub-optimal arm having smaller index than optimal arm do not satisfy Eq. (7), hence it can not be in set $\mathcal{B}_t$. Therefore, the smallest arm of set $\mathcal{B}_t$ is the optimal arm.

**Theorem 1.** *The set $\mathcal{P}_{CWD}$ is maximal learnable.*

The proof establishes that under the CWD property, there exists a 'sound' arm selection policy that identifies the optimal arm for each context. The sound policy only uses conditional disagreement probabilities between pairs of arms that can be estimated from the feedback of arms.

## 3  Parameterization of Pairwise Disagreement Probability

Since the number of contexts could be much larger (can be infinite) than the learning horizon, in stochastic contextual bandits, a correlation structure is assumed between the reward (loss) and the contexts (Auer, 2002; Li et al., 2010, 2017). It is often realized via parameterization of the arms such that expected rewards (or losses) observed from an arm depend on the unknown parameter. In our setting, we cannot observe a loss for any arm. Hence parameterization of an expected loss of the arms is not useful. However, we can obtain feedback of two arms for a given context and can compare them. For example, we can check whether two arms' feedback agrees or disagrees for a given context. Thus, we assume a correlation structure on the disagreement probability for a pair of arms across the contexts and parameterize it using generalized linear models. For $i < j$ and context $x_t$, the disagreement probability for $(i, j)$ pair of arms is given via a function $\mu$ as follows:

$$\mathbb{P}\left\{ Y_t^i \neq Y_t^j | X = x_t \right\} = \mu(\Phi_{ij}(x_t)^\top \theta_{ij}^\star), \tag{8}$$

where $x_t \in \mathbb{R}^d$, $\Phi_{ij} : \mathbb{R}^d \to \mathbb{R}^{d'}$ is a feature map for some $d' \geq d$,[2] and $\theta_{ij}^\star \in \mathbb{R}^{d'}$ is the unknown parameter for $(i, j)$ pair.

We assume the following assumptions on context distribution $\nu$ and function $\mu$, which is standard in the GLM bandit literature (Filippi et al., 2010; Li et al., 2017):

**Assumption 1** (GLM).    • *For all $x \in \mathcal{X}$ and $(i, j)$ pairs, $\|\Phi_{ij}(x)\|_2 \leq 1$.*

- $\kappa \doteq \inf_{\|x\|_2 \leq 1, \|\theta - \theta_{ij}^\star\|_2 \leq 1} \dot{\mu}(\Phi_{ij}(x)^\top \theta) > 0$ *for all $(i, j)$ pairs.*

- *There exists a constant $\lambda_\Sigma > 0$ such that $\lambda_{min} \left( \mathbb{E}\left[ \Phi_{ij}(X)\Phi_{ij}(X)^\top \right] \right) \geq \lambda_\Sigma$ for all $(i, j)$ pairs.*

- *The function $\mu : \mathbb{R} \to [0, 1]$ is continuously differentiable and Lipschitz with constant $k_\mu$.*

For our setting, the function $\mu$ is defined as $\mu(z) = 1/(1 + e^{-z})$, which is the logistic function. The logistic function is widely used function for binary classification model and has $k_\mu \leq 1/4$.

In contextual USS setup, we can compare the arms' feedback and check whether they agree or not for a given context. These binary observations (agree or disagree) can be treated as noisy samples of the disagreement probability. The noise in the binary observation obtained by comparing the feedback of $(i, j)$ pair of arms in round $t$, is given by

$$\varepsilon_{ij}^{(t)} = \begin{cases} 1 - \mu(\Phi_{ij}(x_t)^\top \theta_{ij}^\star), & \text{with probability } \mu(\Phi_{ij}(x_t)^\top \theta_{ij}^\star) \\ -\mu(\Phi_{ij}(x_t)^\top \theta_{ij}^\star), & \text{with probability } \left( 1 - \mu(\Phi_{ij}(x_t)^\top \theta_{ij}^\star) \right) \end{cases}$$

where $\varepsilon_{ij}^{(t)}$ is $\mathcal{F}_t$-measurable with $\mathbb{E}\left[\varepsilon_{ij}^{(t)}|\mathcal{F}_t\right] = 0$. Here $\mathcal{F}_t$ denotes sigma algebra generated by history $\left\{\left(X_s, I_s, \{Y_s^i\}_{i\in[I_s]}\right)\right\}_{s\in[t]}$ till time $t$. Since $\varepsilon_{ij}^{(t)}$ is a zero-mean shifted Bernoulli random variable, $\varepsilon_{ij}^{(t)}$ satisfies the following sub-Gaussian condition with parameter $\sigma \in (0,1)$:

$$\mathbb{E}\left[\exp(\lambda\varepsilon_{ij}^{(t)})|\mathcal{F}_t\right] \leq \exp\left(\frac{\lambda^2\sigma^2}{2}\right), \quad \forall \lambda \in \mathbb{R}.$$

Let $d_{ij}(t) \doteq \mathbb{1}_{\{Y_t^i \neq Y_t^j | X=x_t\}}$ be the disagreement indicator for a context $x_t$ and $S_{ij}^t$ be the set of indices of contexts for which disagreements are observed for $(i,j)$ pair of arms till round $t$. In round $t$, we estimate $\theta_{ij}^\star$, denoted by $\hat{\theta}_{ij}^t$, using the following equation adapted from the maximum likelihood estimator (MLE) used for GLM bandits (Filippi et al., 2010; Li et al., 2017):

$$\sum_{s\in S_{ij}^t} \left(d_{ij}(s) - \mu(\Phi_{ij}(x_s)^\top\theta)\right) \Phi_{ij}(x_s) = 0. \tag{9}$$

In the next section, we develop an algorithm that exploits Lemma 2 for selecting the optimal arm to each context. The algorithm replaces the terms $p_{ij}^{(t)}$ in Lemma 2 by their optimistic estimates.

## 4  Algorithm for Contextual USS: USS-PD

Our algorithm, named USS-PD, is based on the *optimism-in-the-face-of-uncertainty* (OFU) principle. USS-PD works as follows: It takes $\delta$ and $m$ as inputs, where $\delta$ is the confidence in the estimated parameters and used for computing confidence bound for $\theta_{ij}^\star$ as given by Lemma 4. The choice of $m$ ensures that with probability at least $(1-\delta)$, the sample correlation matrix $V_{ij}^t = \sum_{s\in S_{ij}^t} \Phi_{ij}(x_s)\Phi_{ij}(x_s)^\top$ for each $(i,j)$ pair where $i < j$, is invertible. A high probability upper bound on $m$ is computed using Lemma 3. The algorithm collects feedback from all arms by selecting the arm $K$ irrespective of the context received for first $m$ rounds. After $m$ rounds, the sample correlation matrix and the estimate of $\theta_{ij}^\star$ are computed for each $(i,j)$ pair where $i < j$.

For $t > m$, the learner receives a context $x_t$ and plays the arm $i = 1$ and then observe its feedback. For each $(i,j)$ pair and context $x_t$, the upper bound on disagreement probability $\tilde{p}_{ij}^{(t)}$ is computed using $\hat{\theta}_{ij}^t$ and confidence bonus $\alpha_{ij}^t \|\Phi_{ij}(x_t)\|_{(V_{ij}^t)^{-1}}$. Here the notation $\|x\|_A^2 \doteq x^\top A x$ denotes the weighted $l_2$-norm of vector $x \in \mathbb{R}^d$ with respect to a positive definite matrix $A \in \mathbb{R}^{d\times d}$. The confidence bonus has two terms. The first term $\alpha_{ij}^t$ is a slowly increasing function in $t$ whose value is specified in Lemma 4, and the second term $\|\Phi_{ij}(x_t)\|_{(V_{ij}^t)^{-1}}$ decreases to zero as $t$ increases.

---

**USS-PD** Algorithm for Contextual USS using Pairwise Disagreement

1: **Input:** Tuning parameters: $\delta \in (0,1)$ and $m > 0$
2: Select arm $K$ for first $m$ contexts
3: $\forall i < j \leq K$ : set $V_{ij}^m \leftarrow \sum_{t=1}^m \Phi_{ij}(x_t)\Phi_{ij}(x_t)^\top$ and update $\hat{\theta}_{ij}^m$ by solving Eq. (9)
4: **for** $t = m+1, m+2, \dots$ **do**
5:     Receive context $x_t$. Set $i = 1$ and $I_t = 0$
6:     **do**
7:         Play arm $i$
8:         $\forall j \in [i+1, K]$ : compute $\tilde{p}_{ij}^{(t)} \leftarrow \mu\left(\Phi_{ij}(x_t)^\top\hat{\theta}_{ij}^{t-1} + \alpha_{ij}^{t-1}\|\Phi_{ij}(x_t)\|_{(V_{ij}^{t-1})^{-1}}\right)$
9:         If $\forall j \in [i+1, K]$ : $C_j - C_i > \tilde{p}_{ij}^{(t)}$ or $i = K$ then set $I_t = i$ else set $i = i+1$
10:     **while** $I_t = 0$
11:     Select arm $I_t$ and observe $Y_t^1, Y_t^2, \dots, Y_t^{I_t}$
12:     $\forall i < j \leq I_t$ : update $V_{ij}^t \leftarrow V_{ij}^{t-1} + \Phi_{ij}(x_t)\Phi_{ij}(x_t)^\top$ and $\hat{\theta}_{ij}^t$ by solving Eq. (9)
13: **end for**

---

After computing $\tilde{p}_{ij}^{(t)}$, the algorithm checks whether the arm $i$ is the best arm using Eq. (7) with $\tilde{p}_{ij}^{(t)}$ in place of $p_{ij}^{(t)}$. If the arm $i$ is not the best, then the algorithm plays the next arm, and then the same process is repeated. If the arm $i$ is the best arm for context, then the algorithm stops at that arm with $I_t = i$ for that context. After selecting arm $I_t$, the feedback from arms $1, \ldots, I_t$ are observed. After that, the values of $V_{ij}^t$ are updated, and $\hat{\theta}_{ij}^t$ are re-estimated. The same process is repeated for subsequent contexts.

**Remark 1.** *GLM bandits are well studied but require reward or loss information. In the USS setup, loss of selected arm can not be observed; hence finding the optimal arm is challenging. Due to binary disagreement, USS-PD uses the MLE estimator for $\theta_{ij}^\star$ as used in GLM bandits (Filippi et al., 2010; Li et al., 2017). However, the feedback structure and the way arms are selected in the USS setup differ from that in the GLM bandits. Further, our analysis needs carefully connecting the regret with the bad events that make USS-PD selects non-optimal arms.*

**Remark 2.** *We force the algorithm to explore until the correlation matrix $V_{ij}^t$ is invertible for all $(i, j)$ pairs. The invertibility can also be ensured by adding a regularization term (Abbasi-Yadkori et al., 2011; Zhang et al., 2016; Jun et al., 2017) to avoid forced exploration. However, the analysis of USS-PD with regularization term still required to the non-regularized part of the sample correlation matrix becomes invertible. See the supplementary material for the algorithm and its analysis.*

## 4.1 Regret Analysis of USS-PD

The following definition is useful in our regret analysis.

**Definition 2** (Arm Preference ($\succ_t$)). *USS-PD prefers an arm $i$ over $j$ for context $x_t$ if*

$$
i \succ_t j \doteq \begin{cases} C_i - C_j < \tilde{p}_{ji}^{(t)}, & \text{if } j < i \qquad\qquad (10a) \\ C_j - C_i > \tilde{p}_{ij}^{(t)}, & \text{if } j > i. \qquad\qquad (10b) \end{cases}
$$

Our next result bounds the number of disagreement observations required from a pair of arms say $(i, j)$, such that the smallest eigenvalue of its sample correlation matrix $V_{ij}$ matrices is larger than a fixed value. This result uses the standard results from random matrix theory (Vershynin, 2012).

**Lemma 3.** *Let $V_{ij}^t = \sum_{s \in S_{ij}^t} \Phi_{ij}(x_s)\Phi_{ij}(x_s)^\top$, $\Sigma_{ij} = \mathbb{E}\left[\Phi_{ij}(X)\Phi_{ij}(X)^\top\right]$, $\Psi$ and $\delta \in (0,1)$ be two positive constants. Then, there exist positive universal constants $C_1$ and $C_2$ such that the minimum eigenvalue of $\lambda_{min}(V_{ij}^t) \geq \Psi$ with probability at least $1 - 2\delta/K^2$, iff*

$$
|S_{ij}^t| \geq \left( \frac{C_1\sqrt{d'} + C_2\sqrt{\log(K^2/2\delta)}}{\lambda_{min}(\Sigma_{ij})} \right)^2 + \frac{2\Psi}{\lambda_{min}(\Sigma_{ij})}.
$$

The next result is adapted to our setting from the confidence bounds for maximum likelihood estimator used in GLM bandits (Li et al., 2017).

**Lemma 4** (Confidence Ellipsoid). *Let $m$ be such that $\lambda_{min}(V_{ij}^{m+1}) \geq 1$ for any pair $(i, j)$. Then the following event holds with probability at least $1 - 2\delta/K^2$ for USS-PD:*

$$
\left\| \hat{\theta}_{ij}^t - \theta_{ij}^\star \right\|_{V_{ij}^t} \leq \alpha_{ij}^t, \ \forall t > m
$$

*where $\alpha_{ij}^t = \frac{2\sigma}{\kappa}\sqrt{\frac{d'}{2}\log\left(1 + \frac{2t}{d'}\right) + \log\left(\frac{K^2}{2\delta}\right)}$.*

The regret analysis of GLM bandits hinges on bounding the instantaneous regret in each round, which is tied to the estimation error of the GLM parameters. Due to the unsupervised setting and cascade structure, this way of bounding regret does not work in our setup. Our analysis goes by bounding the number of pulls of the sub-optimal arms. However, unlike standard bandits, we have to distinguish whether the sub-optimal arm pulled by USS-PD is on the 'left' or 'right' of the optimal arm in the cascade. It requires our analysis to handle both the cases carefully. Since USS-PD uses a similar MLE estimator for parameter estimation as in GLM bandits, we only adapt their asymptotic normality results. Our next results give conditions when USS-PD prefers a sub-optimal arm for a context.

**Lemma 5.** *Let $\theta \in \Theta_{\mathrm{CWD}}$. Then USS-PD prefers any sub-optimal arm $l < i_t^\star$ for context $x_t$ with probability at most $\delta/2$.*

**Lemma 6.** *Let $\theta \in \Theta_{\mathrm{CWD}}$. If USS-PD prefers a sub-optimal arm $h > i_t^\star$ for context $x_t$ then*

$$2k_\mu \alpha_{i_t^\star h}^t > \xi_{i_t^\star h}(x_t)\sqrt{\lambda_{min}(V_{i_t^\star h}^t)}.$$

*where $\xi_{i_t^\star h} = C_h - C_{i_t^\star} - p_{i_t^\star h}^{(t)}$ and $\alpha_{ij}^t$ is given by Lemma 4.*

Let $m \doteq C\lambda_\Sigma^{-2}\left(d' + \log(k^2/2\delta)\right) + 2\lambda_\Sigma^{-1}$, where $C > 0$ is the universal constant and $R_{max} \doteq \max_{i \in [K], x \in \mathcal{X}}\left[C_i + \gamma_i(x) - (C_{i^\star} + \gamma_{i^\star}(x))\right]$, where $i^\star$ is the optimal arm for context $x$. Now we state the regret upper bound of USS-PD.

**Theorem 2** (Regret Upper Bound). *Let $\theta \in \Theta_{\mathrm{CWD}}$, $\delta \in (0, 1)$, Assumption 1 holds, and $\xi_h = \min_{t \geq 1}\xi_{i_t^\star h}(x_t)$. Then with probability at least $1 - 2\delta$, the regret of USS-PD for $T > m$ contexts is*

$$\mathfrak{R}_T \leq R_{max}\left[m + \sum_{h=2}^K \left(\left(\frac{C_1\sqrt{d'} + C_2\sqrt{\log\left(\frac{K^2}{2\delta}\right)}}{\lambda_\Sigma}\right)^2 + \frac{16}{\lambda_\Sigma}\right.\right.$$
$$\left.\left.\left(\frac{k_\mu\sigma}{\xi_h\kappa}\right)^2\left(\frac{d'}{2}\log\left(1 + \frac{2T}{d'}\right) + \log\left(\frac{K^2}{2\delta}\right)\right)\right)\right].$$

**Corollary 1.** *Let technical conditions stated in Theorem 2 hold. Then with probability at least $1 - 2\delta$*
$$\mathfrak{R}_T \leq O\left(Kd'\log(T)/\xi^2\right).$$

The regret of USS-PD for instance $\theta \in \Theta_{\mathrm{CWD}}$ is logarithmic in $T$ and grows linearly with $d'$ and $K$. The regret is inversely dependent on the value of $\xi \doteq \min_{h \geq 2}\xi_h$ (measure how well CWD holds), which implies the problem instance with smaller $\xi$ has more regret and vice-versa. The value of $\xi$ is analogous to the minimum sub-optimality gap in the standard Multi-Armed Bandits setting. With a large context set, $\xi$ can be small, and its inverse relation in the regret captures the difficulty of the USS problem.

## 5 Experiment

We evaluate the performance of USS-PD on different problem instances derived from synthetic and real datasets. In our experiments, the data samples are treated as contexts. The labels of contexts are known but are never revealed to the algorithm. We use the labels to train classifiers offline that act as arms. Arm $i$ represents a logistic classifier with trained parameter $\theta_i$. A context (data sample) $x$ is assigned label 1 from the $i$-th classifier with probability $\mu(x^\top\theta_i)$ and label 0 with probability $1 - \mu(x^\top\theta_i)$. The disagreement labels for $(i, j)$ pair is computed using the labels of classifier $i$ and $j$. To satisfy Eq. (8), we use the polynomial kernel of degree two for mapping context into higher-dimensional space. Unlike other kernels, the polynomial kernel uses a well-defined feature map to lift the contexts into fixed, higher-dimensional space. The details of the used problem instances are as follows.

**Synthetic Dataset:** We consider 3-dimensional synthetic dataset with 5000 data samples. Each sample is represented by $x = (x_1, x_2, x_3)$, where the value of $x_j$ is drawn uniformly at random from $(-1, 1)$. A sample $x$ is labeled 0 if the value of $(x_1 + x_1x_2 + x_3^2)$ is negative otherwise it is labeled 1. We train five logistic classifiers on this synthetic dataset by varying the regularization parameter. We then assign a positive cost to each classifier and order them by their increasing cost. We vary the cost of using classifiers to get different problem instances (see details in the supplementary material).

**Real Datasets:** We applied our algorithm on PIMA Indian Diabetes (Kaggle, 2016) dataset. Each sample has 8 features related to the conditions of the patient. We split the features into three subsets and train a logistic classifier on each subset. We associate 1st classifier with the first 6 features as input. These features include patient history/profile. The 2nd classifier, in addition to the 6 features, utilizes the feature on the glucose tolerance test, and the 3rd classifier uses all the previous features and the feature that gives values of insulin test. Due to space constraints, the experiment results on Heart Disease dataset (Detrano, 1998; Dheeru and Karra Taniskidou, 2017) are given in the supplementary material.

## 5.1 Experiments Results

We compare the performance of USS-PD on four problem instances derived from the synthetic dataset. The instances vary based on the cost of arms. All contexts in Instance 1 do not satisfy WD property; hence it suffers linear regret as shown in Fig. 1a. For the remaining instances, we set costs such that the value of $\xi$ increasing from Instance 2 to 4. As expected, the regret decreases from Instance 2 to 4, as seen in Fig. 1a. We also compare USS-PD against an algorithm where the learner receives true labels as feedback. In particular, the learner knows whether the classifier's output is correct or not and can estimate their error rates. We implement this 'supervised' setting by replacing disagreement probability in Eq. (7) with estimated error rates. As expected, the regret with supervision has lower than the USS-PD regret (unsupervised) in Fig. 1b. It is qualitatively interesting because these plots demonstrate that, in typical cases, our unsupervised algorithm can eventually learn to perform as good as an algorithm with knowledge of true labels.

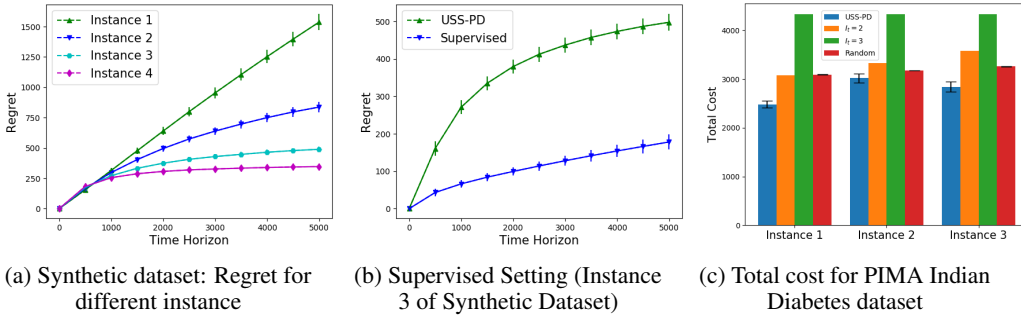

(a) Synthetic dataset: Regret for different instance

(b) Supervised Setting (Instance 3 of Synthetic Dataset)

(c) Total cost for PIMA Indian Diabetes dataset

Figure 1: Performance of USS-PD on different problem instances derived from synthetic and real datasets.

We derive three problem instances from PIMA Indian Diabetes dataset by varying the costs of using classifies. Since all contexts of these problem instances do not satisfy WD property (see details in the supplementary material), we used cumulative total expected cost as a performance measure, where the cumulative total expected cost is given by $\sum_{t=1}^{T}(\gamma_{I_t}(x_t) + C_{I_t})$. We compare the performance of USS-PD with three baseline policies – the first baseline policy uses the third classifier irrespective of contexts, and it is denoted as policy '$I_t = 3$' (plays arm 3 in each round). The second baseline policy uses the second classifier for all contexts, and it is denoted as policy '$I_t = 2$'. The third baseline policy is 'Random,' which selects an arm uniformly at random in each round. In all three problem instances, we observe that USS-PD performs better than the baselines, as shown in Fig. 1c.

We repeat each of the above experiments 100 times, and then the average regret is presented with a 95% confidence interval. The vertical line on each plot shows the confidence interval.

## 6 Conclusion and Future Directions

We studied the unsupervised sequential selection problem with contextual information. It is a partial monitoring stochastic contextual bandit problem, where the loss of an arm can not be inferred from the observed feedback. But one can compare the feedback of two arms to see if they agree or disagree. We modeled the disagreement probability between each pair of the arms as linearly parameterized and developed an algorithm named USS-PD that achieves $O(\log T)$ regret with high probability.

We exploited the contextual information but ignored the inherent side observations due to the arms' cascade structure. By using the side observations, one can tighten the regret bounds. Another interesting future direction is to develop algorithms that decide whether it needs to go further down in the cascade when more information about context is revealed along the cascade.

## 7 Broader Impact

The work considered the unsupervised sequential selection problem with contextual information. While we are not targeting any specific applications, the work has many potential civilian applications.

As usual, these can improve societal conditions, but of course, with any technology, specific deployments need care. However, this is outside of the scope of the present work, which is aimed at improving the basic algorithms and understand the fundamental challenges in this problem setting. Of course, the authors hope that their work will have an altogether positive impact, both by deepening our understanding of challenging sequential decision making under uncertainty and by potential future (careful) applications of the algorithms developed here. Having said this, we do not foresee any immediate negative impact of this work.

**Acknowledgments**

Manjesh K. Hanawal would like to thank the support from INSPIRE faculty fellowships from DST, Government of India, SEED grant (16IRCCSG010) from IIT Bombay, and Early Career Research (ECR) Award from SERB. Csaba Szepesvári gratefully acknowledges funding from the Canada CIFAR AI Chairs Program, Amii, and NSERC. Venkatesh Saligrama would like to acknowledge NSF Grants DMS -2007350 (VS), CCF-2022446, CCF-1955981, and the Data Science Faculty Fellowship from the Rafik B. Hariri Institute.

## Footnotes

[1]In our setup, an arm $i$ could be a classifier that outputs label $Y^i$. The classifier's input could be a context and any combinations of feedback observed from classifiers coming before the arm $i$ in the cascade. For example, consider a case where each arm represents a crowd-sourced worker. After using the first $i$ crowd-sourced workers, the final label can be a function of predicted labels of the first $i$ crowd-sourced workers.

[2]Let $\mathbb{R}^{d_{ij}}$ be the space where Eq. (8) holds for $(i, j)$ pair of arms, and $\Phi_{ij}$ is the feature map that lift $x_t$ from $\mathbb{R}^d$ space to $\mathbb{R}^{d_{ij}}$ space. For simplicity, we take $d' = \max_{\forall i < j \leq K} d_{ij}$.

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
