[Supplementary Material]

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

# Supplementary Material: 'Online Algorithm for Unsupervised Sequential Selection with Contextual Information'

---

## A    Missing proofs from Section 2

### A.1    Proof of Lemma 1

**Lemma 1.** *For any $i$, $j$, and $x_t \in \mathcal{X}$, $\gamma_i(x_t) - \gamma_j(x_t) = p_{ij}^{(t)} - 2\mathbb{P}\left\{Y_t^i = Y_t, Y_t^j \neq Y_t | X = x_t\right\}$.*

*Proof.* Using definition of $\gamma_i(x_t) \doteq \mathbb{P}\left\{Y_t^i \neq Y_t | X = x_t\right\}$, we get

$$\gamma_i(x_t) - \gamma_j(x_t) = \mathbb{P}\left\{Y_t^i \neq Y_t | X = x_t\right\} - \mathbb{P}\left\{Y_t^j \neq Y_t | X = x_t\right\}.$$

As the observed feedback is binary, if $Y_t^i = Y_t^j$ and $Y_t^i \neq Y_t$ then $Y_t^j \neq Y_t$,

$$\gamma_i(x_t) - \gamma_j(x_t) = \mathbb{P}\left\{\cancel{Y_t^i \neq Y_t, Y_t^i = Y_t^j | X = x_t}\right\} + \mathbb{P}\left\{Y_t^i \neq Y_t, Y_t^i \neq Y_t^j | X = x_t\right\}$$
$$- \mathbb{P}\left\{\cancel{Y_t^j \neq Y_t, Y_t^i = Y_t^j | X = x_t}\right\} - \mathbb{P}\left\{Y_t^j \neq Y_t, Y_t^i \neq Y_t^j | X = x_t\right\}.$$

Adding and subtracting $\mathbb{P}\left\{Y_t^i = Y_t, Y_t^i \neq Y_t^j | X = x_t\right\}$,

$$\gamma_i(x_t) - \gamma_j(x_t) = \mathbb{P}\left\{Y_t^i \neq Y_t, Y_t^i \neq Y_t^j | X = x_t\right\} + \mathbb{P}\left\{Y_t^i = Y_t, Y_t^i \neq Y_t^j | X = x_t\right\}$$
$$- \mathbb{P}\left\{Y_t^j \neq Y_t, Y_t^i \neq Y_t^j | X = x_t\right\} - \mathbb{P}\left\{Y_t^i = Y_t, Y_t^i \neq Y_t^j | X = x_t\right\}.$$

If $Y_t^i \neq Y_t^j$ and $Y_t^j \neq Y_t$ then $Y_t^i = Y_t$,

$$\gamma_i(x_t) - \gamma_j(x_t) = \mathbb{P}\left\{Y_t^i \neq Y_t^j | X = x_t\right\} - \mathbb{P}\left\{Y_t^i = Y_t, Y_t^i \neq Y_t^j | X = x_t\right\}$$
$$- \mathbb{P}\left\{Y_t^i = Y_t, Y_t^i \neq Y_t^j | X = x_t\right\}$$
$$= \mathbb{P}\left\{Y_t^i \neq Y_t^j | X = x_t\right\} - 2\mathbb{P}\left\{Y_t^i = Y_t, Y_t^j \neq Y_t | X = x_t\right\}.$$
$$\implies \gamma_i(x_t^i) - \gamma_j(x_t^j) = p_{ij}^{(t)} - 2\mathbb{P}\left\{Y_t^i = Y_t, Y_t^j \neq Y_t | X = x_t\right\}. \qquad \square$$

### A.2    Proof of Lemma 2

**Lemma 2.** *Let $P \in \mathcal{P}_{CWD}$ and $\mathcal{B}_t = \left\{i : \forall j > i, C_j - C_i > p_{ij}^{(t)}\right\} \cup \{K\}$. Then the arm $I_t = \min(\mathcal{B}_t)$ is the optimal arm for a context $x_t$.*

*Proof.* Let $i_t^\star$ be an optimal arm for a context $x_t$. As $p_{ij}^{(t)} = \mathbb{P}\{Y_t^i \neq Y_t^j | X = x_t\}$ and $i_t^\star$ is an optimal arm, we have $\forall j < i_t^\star : C_{i_t^\star} - C_j \leq \mathbb{P}\{Y_t^{i_t^\star} \neq Y_t^j | X = x_t\} \implies C_{i_t^\star} - C_j \not> \mathbb{P}\{Y_t^{i_t^\star} \neq Y_t^j | X = x_t\} \implies \forall j < i_t^\star \notin \mathcal{B}_t$. If any sub-optimal arm $h \in \mathcal{B}_t$ then $h > i_t^\star$ i.e.,

$$\mathcal{B}_t = \{i_t^\star, h_1, \ldots, h_n, K\},$$

where $i_t^\star < h_1 < \cdots < h_n < K$. By construction of set $\mathcal{B}_t$, the minimum indexed arm in set $\mathcal{B}_t$ is only the optimal arm. $\qquad \square$

### A.3 Proof of Theorem 1

We need the following results to proof of Theorem 1.

**Lemma 7.** *Let $i < j$ and $x_t \in \mathcal{X}$ be any context. Assume*

$$C_j - C_i \notin \left( \gamma_i(x_t) - \gamma_j(x_t), \mathbb{P}\left\{Y_t^i \neq Y_t^j | X = x_t\right\} \right]. \tag{11}$$

*Then, $C_j - C_i > \gamma_i(x_t) - \gamma_j(x_t)$ iff $C_j - C_i > \mathbb{P}\left\{Y_t^i \neq Y_t^j | X = x_t\right\}$.*

*Proof.* Assume that $C_j - C_i > \gamma_i(x_t) - \gamma_j(x_t)$. As $C_j - C_i \notin \left( \gamma_i(x_t) - \gamma_j(x_t), \mathbb{P}\left\{Y_t^i \neq Y_t^j | X = x_t\right\} \right]$, we get $C_j - C_i > \mathbb{P}\left\{Y_t^i \neq Y_t^j | X = x_t\right\}$. The proof of other direction follows by noting that $\mathbb{P}\left\{Y_t^i \neq Y_t^j | X = x_t\right\} \geq \gamma_i(x_t) - \gamma_j(x_t)$. $\qquad\square$

**Lemma 8.** *Let $i > j$ and $x_t \in \mathcal{X}$ be any context. Assume*

$$C_i - C_j \notin \left( \gamma_j(x_t) - \gamma_i(x_t), \mathbb{P}\left\{Y_t^i \neq Y_t^j | X = x_t\right\} \right]. \tag{12}$$

*Then, $C_i - C_j \leq \gamma_j(x_t) - \gamma_i(x_t)$ iff $C_j - C_i \leq \mathbb{P}\left\{Y_t^i \neq Y_t^j | X = x_t\right\}$.*

*Proof.* Let $C_i - C_j \leq \gamma_j(x_t) - \gamma_i(x_t)$. As $\gamma_j(x_t) - \gamma_i(x_t) \leq \mathbb{P}\left\{Y_t^i \neq Y_t^j | X = x_t\right\}$, we get $C_i - C_j \leq \mathbb{P}\left\{Y_t^i \neq Y_t^j | X = x_t\right\}$.

The condition $C_i - C_j \leq \mathbb{P}\left\{Y_t^i \neq Y_t^j | X = x_t\right\}$ along with $C_i - C_j \notin \left( \gamma_j(x_t) - \gamma_i(x_t), \mathbb{P}\left\{Y_t^i \neq Y_t^j | X = x_t\right\} \right]$ implies the other direction, i.e., $C_i - C_j \leq \gamma_j(x_t) - \gamma_i(x_t)$. $\qquad\square$

**Lemma 9.** *Let $i_t^\star$ be an optimal arm for a context $x_t$. Any problem instance $P \in \mathcal{P}_{USS}$ is learnable if for every context in $P$ following holds:*

$$\forall j > i_t^\star, \; C_j - C_{i_t^\star} > \mathbb{P}\left\{Y_t^{i_t^\star} \neq Y_t^j | X = x_t\right\}.$$

The proof of Lemma 9 follows from Lemma 7 and Lemma 8. Now we give proof for Theorem 1.

**Theorem 1.** *The set $\mathcal{P}_{CWD}$ is maximal learnable.*

*Proof.* Let $i_t^\star$ be an optimal arm for a context $x_t$. It is enough to prove that any problem instance $P \in \mathcal{P}_{USS}$ is learnable if

$$\forall j > i_t^\star, \; C_j - C_{i_t^\star} > \mathbb{P}\left\{Y_t^{i_t^\star} \neq Y_t^j | X = x_t\right\}. \quad \text{(definition of CWD property)}$$

From Lemma 7 and Lemma 8, if the optimal arm satisfies following conditions,

$$\forall j > i_t^\star, C_j - C_{i_t^\star} \notin \left( \gamma_{i_t^\star}(x_t) - \gamma_j(x_t), \mathbb{P}\left\{Y_t^{i_t^\star} \neq Y_t^j | X = x_t\right\} \right] \text{ and}$$

$$\forall j < i_t^\star, C_{i^\star} - C_j \notin \left( \gamma_j(x_t) - \gamma_{i_t^\star}(x_t), \mathbb{P}\left\{Y_t^{i_t^\star} \neq Y_t^j | X = x_t\right\} \right],$$

then, for $j > i_t^\star, C_j - C_{i_t^\star} > \gamma_{i_t^\star}(x_t) - \gamma_j(x)$ iff $C_j - C_{i_t^\star} > \mathbb{P}\left\{Y_t^{i_t^\star} \neq Y_t^j | X = x_t\right\}$ and for $j < i_t^\star, C_{i^\star} - C_j \leq \gamma_j(x) - \gamma_{i_t^\star}(x_t)$ iff $C_j - C_{i_t^\star} \leq \mathbb{P}\left\{Y_t^{i_t^\star} \neq Y_t^j | X = x_t\right\}$. Hence we can use $\mathbb{P}\left\{Y_t^i \neq Y_t^j | X = x_t\right\}$ as a proxy for $\gamma_{i_t^\star}(x) - \gamma_j(x)$ to make decision about the optimal arm. Now notice that for $j < i_t^\star, C_{i^\star} - C_j \leq \gamma_j(x) - \gamma_{i_t^\star}(x_t)$. Hence,

$$\forall j < i_t^\star, C_{i^\star} - C_j \notin \left( \gamma_j(x_t) - \gamma_{i_t^\star}(x_t), \mathbb{P}\left\{Y_t^{i_t^\star} \neq Y_t^j | X = x_t\right\} \right] \text{ and}$$

$$\forall j > i_t^\star, C_j - C_{i_t^\star} \notin \left( \gamma_{i_t^\star}(x_t) - \gamma_j(x_t), \mathbb{P}\left\{ Y_t^{i_t^\star} \neq Y_t^j | X = x_t \right\} \right] \tag{13}$$

are sufficient for learnability. Note that Eq. (13) is equivalent to

$$\forall j > i_t^\star, \ C_j - C_{i_t^\star} > \mathbb{P}\left\{ Y_t^{i_t^\star} \neq Y_t^j | X = x_t \right\}. \tag{14}$$

Note that if Eq. (14) does not hold, then knowing $\mathbb{P}\left\{ Y_t^{i_t^\star} \neq Y_t^j | X = x_t \right\}$ is not sufficient for finding the optimal arm. $\qquad\square$

### A.4 Regret decomposition when contexts satisfy WD with some known probability

Without knowing the disagreement probability, it is impossible to check whether a context satisfies WD property or not. Hence we consider a case where a context can satisfy WD property with some fixed probability. For such cases, we can decompose the regret into two parts: regret due to the contexts that satisfy WD property and regret due to the contexts that do not satisfy WD property. Note that the regret can be linear due to the contexts that do not satisfy the WD condition.

Our next result gives the upper bound on the regret where the contexts satisfy WD property with a known fixed probability.

**Lemma 10.** *Let $\rho$ be the probability of context that it does not satisfy the* WD *property and $R_{max}$ be the maximum regret incurred for any context. If $\mathfrak{R}_T$ is the regret incurred when all contexts satisfy* WD *property then, the regret incurred when contexts satisfy* WD *with probability $(1 - \rho)$ is given by*

$$\mathfrak{R}'_T \leq (1 - \rho)\mathfrak{R}_T + \rho R_{max} T.$$

*Proof.* Let $\rho$ be the probability of context that it does not satisfy the WD property and $r_t(I_t, i_t^\star)$ be the regret incurred for selecting sub-optimal arm $I_t$ for the context $x_t$. Then the regret can be decomposed into two parts as follows:

$$\mathfrak{R}'_T = \mathbb{E}\left[ \sum_{t=1}^{T} \left[ \mathbb{1}_{\{x_t \text{ satisfies WD}\}} r_t(I_t, i_t^\star) + \mathbb{1}_{\{x_t \text{ does not satisfy WD}\}} r_t(I_t, i_t^\star) \right] \right]$$

$$= \mathbb{E}\left[ \sum_{t=1}^{T} \mathbb{1}_{\{x_t \text{ satisfies WD}\}} r_t(I_t, i_t^\star) \right] + \mathbb{E}\left[ \sum_{t=1}^{T} \mathbb{1}_{\{x_t \text{ does not satisfy WD}\}} r_t(I_t, i_t^\star) \right]$$

$$= \sum_{t=1}^{T} \mathbb{P}\left\{ x_t \text{ satisfies WD} \right\} r_t(I_t, i_t^\star) + \sum_{t=1}^{T} \mathbb{P}\left\{ x_t \text{ does not satisfy WD} \right\} r_t(I_t, i_t^\star). \tag{15}$$

First, we will bound the regret due to the contexts that do not satisfy WD property (second term of Eq. (15)). Note that the context that does not satisfy WD property, the learner can not make the correct decision hence always incurs regret. Since the maximum regret is upper bounded by $R_{max}$, we have

$$\sum_{t=1}^{T} \mathbb{P}\left\{ x_t \text{ does not satisfy WD} \right\} r_t(I_t, i_t^\star) \leq \sum_{t=1}^{T} \mathbb{P}\left\{ x_t \text{ does not satisfy WD} \right\} R_{max}$$

Since $\rho$ is the probability of context that it does not satisfy the WD property, we get

$$= \sum_{t=1}^{T} \rho R_{max}$$

$$\implies \sum_{t=1}^{T} \mathbb{P}\left\{ x_t \text{ does not satisfy WD} \right\} r_t(I_t, i_t^\star) \leq \rho R_{max} T. \tag{16}$$

Now we will bound the regret due to the contexts which satisfy WD property (first term in Eq. (15)). Since any context satisfies WD with $1 - \rho$ probability, we have

$$\sum_{t=1}^{T} \mathbb{P}\left\{ x_t \text{ satisfies WD} \right\} r_t(I_t, i_t^\star) = \sum_{t=1}^{T} (1 - \rho) r_t(I_t, i_t^\star) = (1 - \rho) \sum_{t=1}^{T} r_t(I_t, i_t^\star). \tag{17}$$

By assuming that all contexts are satisfying WD property, we have regret $\mathfrak{R}_T = \sum_{t=1}^{T} r_t(I_t, i_t^\star)$. Using it with Eq. (16) in Eq. (17), we get

$$\mathfrak{R}'_T \leq (1 - \rho)\mathfrak{R}_T + \rho R_{max}T. \qquad \square$$

## B  Missing proofs from Section 4

### B.1  Proof of Lemma 3

**Lemma 3.** *Let $V_{ij}^t = \sum_{s \in S_{ij}^t} \Phi_{ij}(x_s)\Phi_{ij}(x_s)^\top$, $\Sigma_{ij} = \mathbb{E}\left[\Phi_{ij}(X)\Phi_{ij}(X)^\top\right]$, $\Psi$ and $\delta \in (0, 1)$ be two positive constants. Then, there exist positive universal constants $C_1$ and $C_2$ such that the minimum eigenvalue of $\lambda_{min}(V_{ij}^t) \geq \Psi$ with probability at least $1 - 2\delta/K^2$, iff*

$$|S_{ij}^t| \geq \left( \frac{C_1\sqrt{d'} + C_2\sqrt{\log(K^2/2\delta)}}{\lambda_{min}(\Sigma_{ij})} \right)^2 + \frac{2\Psi}{\lambda_{min}(\Sigma_{ij})}.$$

*Proof.* The result is adapted from (Li et al., 2017, Proposition 1), which uses the standard random matrix theory result from (Vershynin, 2012, Theorem 5.39). We need to carefully construct the sample complexity bound for our case as the observations are only observed for a pair of arms. $\square$

The following result is needed to prove Lemma 4.

**Lemma 11.** *Let $\overline{V}_{ij}^t = \lambda I_{d'} + a \sum_{s \in S_{ij}^t} \Phi_{ij}(x_s)\Phi_{ij}(x_s)^\top$ for any $(i, j)$ pair of arms, and $n_{ij}^t = |S_{ij}^t|$. Then*

$$det(\overline{V}_{ij}^t) \leq \left( \lambda + an_{ij}^t/d' \right)^{d'}.$$

*Proof.* The proof is adapted from Lemma 10 of Abbasi-Yadkori et al. (2011). By using inequality of arithmetic and geometric means, we have $det(\overline{V}_{ij}^t) \leq (trace(\overline{V}_{ij}^t)/d')^{d'}$. As the trace of matrix is a linear mapping i.e. $trace(A + B) = trace(A) + trace(B)$, hence, we get

$$trace(\overline{V}_{ij}^t) = trace(\lambda I_{d'}) + a \sum_{s \in S_{ij}^t} trace\left(\Phi_{ij}(x_s)\Phi_{ij}(x_s)^\top\right)$$

$$= \lambda d' + a \sum_{s \in S_{ij}^t} \|\Phi_{ij}(x)\|_2^2$$

$$\leq \lambda d' + an_{ij}^t. \qquad \left(\text{as } \|\Phi_{ij}(x)\|_2 \leq 1 \text{ and } n_{ij}^t = |S_{ij}^t|\right)$$

Using upper bound of $trace(\overline{V}_{ij}^t)$ for bounding $det(\overline{V}_{ij}^t)$, we get

$$det(\overline{V}_{ij}^t) \leq (trace(\overline{V}_{ij}^t)/d')^{d'} \leq \left((\lambda d' + an_{ij}^t)/d'\right)^{d'} \leq \left(\lambda + an_{ij}^t/d'\right)^{d'}. \qquad \square$$

### B.2  Proof of Lemma 4

**Lemma 4** (Confidence Ellipsoid). *Let $m$ be such that $\lambda_{min}(V_{ij}^{m+1}) \geq 1$ for any pair $(i, j)$. Then the following event holds with probability at least $1 - 2\delta/K^2$ for USS-PD:*

$$\left\| \hat{\theta}_{ij}^t - \theta_{ij}^\star \right\|_{V_{ij}^t} \leq \alpha_{ij}^t, \ \forall t > m$$

*where $\alpha_{ij}^t = \frac{2\sigma}{\kappa}\sqrt{\frac{d'}{2}\log\left(1 + \frac{2t}{d'}\right) + \log\left(\frac{K^2}{2\delta}\right)}$.*

*Proof.* Let $\overline{V}_{ij}^t = \lambda I_{d'} + V_{ij}^t$. If Eq. (9) is used for estimation of unknown parameter $\theta_{ij}^\star$ then by using Eq. (26) and Lemma 8 of Li et al. (2017) with $\lambda_{min}(V_{ij}^{m+1}) \geq 1$, we have

$$\left\| \hat{\theta}_{ij}^t - \theta_{ij}^\star \right\|_{V_{ij}^t} \leq \frac{1}{\kappa}\left\| \sum_{s \in S_{ij}^t} \varepsilon_s \Phi_{ij}(x_s) \right\|_{(V_{ij}^t)^{-1}} \leq \frac{(1 - \lambda)^{\frac{-1}{2}}}{\kappa}\left\| \sum_{s \in S_{ij}^t} \varepsilon_s \Phi_{ij}(x_s) \right\|_{(\overline{V}_{ij}^t)^{-1}}.$$

Using upper bound of $\left\|\sum_{s \in S_{ij}^t} \varepsilon_s \Phi_{ij}(x_s)\right\|_{(\overline{V}_{ij}^t)^{-1}}$ as given in Theorem 1 of Abbasi-Yadkori et al. (2011) where $\varepsilon_s$ is $\sigma-$subGaussian random variable, the following inequality holds with at least probability $1 - 2\delta/K^2$

$$\leq \frac{(1-\lambda)^{\frac{-1}{2}}}{\kappa} \sqrt{2\sigma^2 \log\left(\frac{det(\overline{V}_{ij}^t)^{1/2} det(\lambda \mathrm{I}_{d'})^{-1/2}}{2\delta/K^2}\right)}$$

$$= \frac{\sigma(1-\lambda)^{\frac{-1}{2}}}{\kappa} \sqrt{2\log\left(\frac{det(\overline{V}_{ij}^t)}{det(\lambda \mathrm{I}_{d'})}\right)^{\frac{1}{2}} + 2\log\left(\frac{K^2}{2\delta}\right)}.$$

Upper bounding $det(\overline{V}_{ij}^t)$ with $a = 1$, $\lambda = 1/2$, and $n_{ij}^t \leq t$ by using Lemma 11, we get

$$\implies \left\|\hat{\theta}_{ij}^t - \theta_{ij}^\star\right\|_{V_{ij}^t} \leq \frac{2\sigma}{\kappa} \sqrt{\frac{d'}{2} \log\left(1 + \frac{2n_{ij}^t}{d'}\right) + \log\left(\frac{K^2}{2\delta}\right)}$$

$$\leq \frac{2\sigma}{\kappa} \sqrt{\frac{d'}{2} \log\left(1 + \frac{2t}{d'}\right) + \log\left(\frac{K^2}{2\delta}\right)}. \qquad \square$$

## B.3  Proof of Lemma 5

**Lemma 5.** *Let $\theta \in \Theta_{\mathrm{CWD}}$. Then USS-PD prefers any sub-optimal arm $l < i_t^\star$ for context $x_t$ with probability at most $\delta/2$.*

*Proof.* If sub-optimal arm $l < i_t^\star$ is preferred by USS-PD then using Eq. (10b), we get

$$\mathbb{1}_{\{l \succ_t i_t^\star, i_t^\star = i\}} = \mathbb{1}_{\left\{C_i - C_l > \tilde{p}_{li}^{(t)}, I_t = l, i_t^\star = i\right\}}$$

$$\leq \mathbb{1}_{\left\{C_i - C_l > \tilde{p}_{li}^{(t)}\right\}}. \qquad \text{(as } A \cap B \cap C \subseteq A\text{)}$$

Using $C_i - C_l = p_{li}(x_t) - \xi_{li}(x_t)$ for $l < i$, we have

$$\implies \mathbb{1}_{\{l \succ_t i_t^\star, i_t^\star = i\}} = \mathbb{1}_{\left\{p_{li}(x_t) - \xi_{li}(x_t) > \tilde{p}_{li}^{(t)}\right\}} = \mathbb{1}_{\left\{p_{li}(x_t) - \tilde{p}_{li}^{(t)} > \xi_{li}(x_t)\right\}}.$$

Using definition of $p_{li}(x_t)$ and $\tilde{p}_{li}^{(t)}$,

$$\implies \mathbb{1}_{\{l \succ_t i_t^\star, i_t^\star = i\}} = \mathbb{1}_{\left\{\mu(\Phi_{li}(x_t)^\top \theta_{li}^\star) - \mu\left(\Phi_{li}(x_t)^\top \hat{\theta}_{li}^t + \alpha_{li}^t \|\Phi_{li}(x_t)\|_{(V_{li}^t)^{-1}}\right) > \xi_{li}(x_t)\right\}}.$$

Since $\mu(\cdot)$ is an increasing function and using $\alpha_{li}^t$ as defined in Lemma 4, $\mu\left(\Phi_{li}(x_t)^\top \hat{\theta}_{li}^t + \alpha_{li}^t \|\Phi_{li}(x_t)\|_{(V_{li}^t)^{-1}}\right)$ is the upper bound on $\mu(\Phi_{li}(x_t)^\top \theta_{li}^\star)$ for all $(l, i)$ pairs with probability at least $1 - \delta/2$. We show it as follows:

$$\Phi_{li}(x_t)^\top \theta_{li}^\star = \Phi_{li}(x_t)^\top \hat{\theta}_{li}^t + \Phi_{li}(x_t)^\top (\theta_{li}^\star - \hat{\theta}_{li}^t)$$

$$= \Phi_{li}(x_t)^\top \hat{\theta}_{li}^t + \|\Phi_{li}(x_t)\|_{(V_{li}^t)^{-1}} \left\|\theta_{li}^\star - \hat{\theta}_{li}^t\right\|_{V_{li}^t}$$

$$\implies \Phi_{li}(x_t)^\top \theta_{li}^\star \leq \Phi_{li}(x_t)^\top \hat{\theta}_{li}^t + \alpha_{li}^t \|\Phi_{li}(x_t)\|_{(V_{li}^t)^{-1}}. \quad \left(\text{using } \left\|\theta_{li}^\star - \hat{\theta}_{li}^t\right\|_{V_{li}^t} \leq \alpha_{li}^t\right)$$

Since $\mu(\cdot)$ is an increasing function,

$$\implies \mu(\Phi_{li}(x_t)^\top \theta_{li}^\star) \leq \mu\left(\Phi_{li}(x_t)^\top \hat{\theta}_{li}^t + \alpha_{li}^t \|\Phi_{li}(x_t)\|_{(V_{li}^t)^{-1}}\right).$$

Hence, any sub-optimal arm smaller than the optimal arm is selected by USS-PD with probability at most $\delta/2$. It completes the proof of the lemma. $\qquad \square$

## B.4 Proof of Lemma 6

**Lemma 6.** *Let $\theta \in \Theta_{\mathrm{CWD}}$. If USS-PD prefers a sub-optimal arm $h > i_t^\star$ for context $x_t$ then*

$$2k_\mu \alpha_{i_t^\star h}^t > \xi_{i_t^\star h}(x_t)\sqrt{\lambda_{min}(V_{i_t^\star h}^t)}.$$

*where $\xi_{i_t^\star h} = C_h - C_{i_t^\star} - p_{i_t^\star h}^{(t)}$ and $\alpha_{ij}^t$ is given by Lemma 4.*

*Proof.* If sub-optimal arm $h > i_t^\star$ is preferred by USS-PD then using Eq. (10a), we get

$$\mathbb{1}_{\{h \succ_t i, i_t^\star = i\}} = \mathbb{1}_{\left\{C_h - C_i < \tilde{p}_{ih}^{(t)}, h \succ_t i_t^\star, i_t^\star = i\right\}}$$

$$\leq \mathbb{1}_{\left\{C_h - C_i < \tilde{p}_{ih}^{(t)}\right\}}. \quad (\text{as } A \cap B \cap C \subseteq A)$$

Using $C_h - C_i = p_{ih}(x_t) + \xi_{ih}(x_t)$ for $h > i$, we get

$$\implies \mathbb{1}_{\{h \succ_t i, i_t^\star = i\}} = \mathbb{1}_{\left\{p_{ih}(x_t) + \xi_{ih}(x_t) < \tilde{p}_{ih}^{(t)}\right\}} = \mathbb{1}_{\left\{\tilde{p}_{ih}^{(t)} - p_{ih}(x_t) > \xi_{ih}(x_t)\right\}}.$$

Using definition of $p_{ih}(x_t)$ and $\tilde{p}_{ih}^{(t)}$,

$$\implies \mathbb{1}_{\{h \succ_t i, i_t^\star = i\}} = \mathbb{1}_{\left\{\mu\left(\Phi_{ih}(x_t)^\top \theta_{ih}^\star + \alpha_{ih}^t \|\Phi_{ih}(x_t)\|_{(V_{ih}^t)^{-1}}\right) - \mu(\Phi_{ih}(x_t)^\top \hat{\theta}_{ih}^t) > \xi_{ih}(x_t)\right\}}.$$

As $\mu$ is Lipschitz, $|\mu(z_1) - \mu(z_2)| \leq k_\mu|z_1 - z_2|$ where $k_\mu$ is Lipschitz constant, we have

$$\leq \mathbb{1}_{\left\{k_\mu|\Phi_{ih}(x_t)^\top \theta_{ih}^\star + \alpha_{ih}^t\|\Phi_{ih}(x_t)\|_{(V_{ih}^t)^{-1}} - \Phi_{ih}(x_t)^\top \hat{\theta}_{ih}^t| > \xi_{ih}(x_t)\right\}}$$

$$\leq \mathbb{1}_{\left\{k_\mu|\Phi_{ih}(x_t)^\top \theta_{ih}^\star - \Phi_{ih}(x_t)^\top \hat{\theta}_{ih}^t| + k_\mu \alpha_{ih}^t\|\Phi_{ih}(x_t)\|_{(V_{ih}^t)^{-1}} > \xi_{ih}(x_t)\right\}}$$

$$= \mathbb{1}_{\left\{k_\mu|\Phi_{ih}(x_t)^\top (\theta_{ih}^\star - \hat{\theta}_{ih}^t)| + k_\mu \alpha_{ih}^t\|\Phi_{ih}(x_t)\|_{(V_{ih}^t)^{-1}} > \xi_{ih}(x_t)\right\}}.$$

Using Cauchy-Schwartz inequality and $\|x\|_A^2 = x^\top A x$, we get

$$\leq \mathbb{1}_{\left\{k_\mu\|\Phi_{ih}\|_{(V_{ih}^t)^{-1}}\left\|\theta_{ih}^\star - \hat{\theta}_{ih}^t\right\|_{V_{ih}^t} + k_\mu \alpha_{ih}^t\|\Phi_{ih}(x_t)\|_{(V_{ih}^t)^{-1}} > \xi_{ih}(x_t)\right\}}.$$

As $\left\|\theta_{ih}^\star - \hat{\theta}_{ih}^t\right\|_{V_{ih}^t} \leq \alpha_{ih}^t$, we get

$$\leq \mathbb{1}_{\left\{k_\mu \alpha_{ih}^t\|\Phi_{ih}(x_t)\|_{(V_{ih}^t)^{-1}} + k_\mu \alpha_{ih}^t\|\Phi_{ih}(x_t)\|_{(V_{ih}^t)^{-1}} > \xi_{ih}(x_t)\right\}}$$

$$= \mathbb{1}_{\left\{2k_\mu \alpha_{ih}^t\|\Phi_{ih}(x_t)\|_{(V_{ih}^t)^{-1}} > \xi_{ih}(x_t)\right\}}.$$

As $\|\Phi_{ih}(x_t)\|_{(V_{ih}^t)^{-1}} \leq \|\Phi_{ih}(x_t)\|_2 / \sqrt{\lambda_{min}(V_{ih}^t)}$ where $\lambda_{min}(V_{ih}^t)$ is the smallest eigenvalue of matrix $V_{ih}^t$ and $\|\Phi_{ih}(x_t)\|_2 \leq 1$, we get

$$\implies \mathbb{1}_{\{h \succ_t i, i_t^\star = i\}} \leq \mathbb{1}_{\left\{2k_\mu \alpha_{ih}^t > \xi_{ih}(x_t)\sqrt{\lambda_{min}(V_{ih}^t)}\right\}}. \tag{18}$$

The event on LHS is subset of event of RHS in Eq. (18). By changing $i$ to $i_t^\star$ completes the proof of the lemma. $\square$

## B.5 Proof of Theorem 2

**Theorem 2** (Regret Upper Bound). *Let $\theta \in \Theta_{\mathrm{CWD}}$, $\delta \in (0, 1)$, Assumption 1 holds, and $\xi_h = \min_{t \geq 1} \xi_{i_t^\star h}(x_t)$. Then with probability at least $1 - 2\delta$, the regret of USS-PD for $T > m$ contexts is*

$$\mathfrak{R}_T \leq R_{max}\left[m + \sum_{h=2}^{K}\left(\left(\frac{C_1\sqrt{d'} + C_2\sqrt{\log\left(\frac{K^2}{2\delta}\right)}}{\lambda_\Sigma}\right)^2 + \frac{16}{\lambda_\Sigma}\right.\right.$$

$$\left.\left.\left(\frac{k_\mu \sigma}{\xi_h \kappa}\right)^2\left(\frac{d'}{2}\log\left(1 + \frac{2T}{d'}\right) + \log\left(\frac{K^2}{2\delta}\right)\right)\right)\right].$$

*Proof.* The regret for $T$ rounds in the Contextual USS problem is given by

$$\mathfrak{R}_T = \sum_{t=1}^{T} \left( C_{I_t} + \gamma_{I_t}(x_t) - (C_{i_t^\star} + \gamma_{i_t^\star}(x_t)) \right).$$

As $R_{max}$ denote the maximum regret incurred for any context, we get

$$\mathfrak{R}_T \leq R_{max} \sum_{t=1}^{T} \mathbb{1}_{\{I_t \neq i_t^\star\}}. \tag{19}$$

As $\mathbb{1}_{\{I_t \neq i_t^\star\}}$ has two random quantities $I_t$ and $i_t^\star$, we can re-write it as follows:

$$\mathbb{1}_{\{I_t \neq i_t^\star\}} = \sum_{l < i} \mathbb{1}_{\{I_t = l, i_t^\star = i\}} + \sum_{h' > i} \mathbb{1}_{\{I_t = h', i_t^\star = i\}}.$$

Note that if USS-PD selects $l < i_t^\star$ then $l$ must be preferred over $i_t^\star$ whereas if $h' > i_t^\star$ is selected then there exists an arm $h > i_t^\star$ which is preferred over $i_t^\star$. Hence, we have

$$\mathbb{1}_{\{I_t \neq i_t^\star\}} = \sum_{l < i} \mathbb{1}_{\{l \succ_t i_t^\star, i_t^\star = i\}} + \sum_{h' > i} \mathbb{1}_{\{I_t = h', h \succ_t i_t^\star, i_t^\star = i\}}$$

$$\leq \sum_{l < i} \mathbb{1}_{\{l \succ_t i_t^\star, i_t^\star = i\}} + \sum_{h > i} \mathbb{1}_{\{h \succ_t i_t^\star, i_t^\star = i\}}. \tag{20}$$

Using above bound in Eq. (19), we get

$$\mathfrak{R}_T \leq R_{max} \sum_{t=1}^{T} \left[ \sum_{l < i} \mathbb{1}_{\{l \succ_t i_t^\star, i_t^\star = i\}} + \sum_{h > i} \mathbb{1}_{\{h \succ_t i_t^\star, i_t^\star = i\}} \right].$$

From Lemma 5, $\mathbb{1}_{\{l \succ_t i_t^\star, i_t^\star = i\}} = 0$ for any $l < i$ with probability at least $1 - \delta/2$, then the regret becomes

$$\mathfrak{R}_T \leq R_{max} \sum_{t=1}^{T} \sum_{h > i} \mathbb{1}_{\{h \succ_t i_t^\star, i_t^\star = i\}} = R_{max} \sum_{h > i} \sum_{t=1}^{T} \mathbb{1}_{\{h \succ_t i_t^\star, i_t^\star = i\}} \leq R_{max} \sum_{h=2}^{K} \sum_{t=1}^{T} \mathbb{1}_{\{h \succ_t i_t^\star, i_t^\star < h\}}.$$

Note that $\alpha_{ih}^t$ is slowly increasing value with $t$ that implies $\alpha_{ih}^t \leq \alpha_{ih}^T$ for all $t \leq T$. Using Lemma 3 with $\Psi = \left( \frac{2k_\mu \alpha_{ih}^T}{\xi_{ih}} \right)^2$, $\Sigma_{ih} = \mathbb{E}\left[ \Phi_{ih}(X_s) \Phi_{ih}(X_s)^T \right]$ where $s \in S_{ih}^t$, after

$$n_{ih}^T \doteq \left( \frac{C_1 \sqrt{d'} + C_2 \sqrt{\log(K^2/2\delta)}}{\lambda_{min}(\Sigma_{ih})} \right)^2 + \frac{2}{\lambda_{min}(\Sigma_{ih})} \left( \frac{2k_\mu \alpha_{ih}^T}{\xi_{ih}} \right)^2$$

observations for arm pair $(i, h)$ the $\lambda_{min}(V_{ih}^t) \geq \left( \frac{2k_\mu \alpha_{ih}^T}{\xi_{ih}} \right)^2$ with probability at least $1 - 2\delta/K^2$. Therefore, after having $n_{ih}^T$ observations, the sub-optimal arm $h(> i)$ will not be preferred over optimal arm $i$ with probability at least $1 - 2\delta/K^2$. Therefore, with probability at least $1 - 2\delta/K^2$, following equations also hold

$$\mathbb{1}_{\left\{ I_t = h, i_t^\star = i, |S_{ih}^t| \geq n_{ih}^T \right\}} = 0 \implies \sum_{t=1}^{T} \mathbb{1}_{\{I_t = h, i_t^\star = i, h > i\}} \leq n_{ih}^T.$$

Due to the problem structure, whenever an arm $h$ is selected, disagreement labels for all arm pair $(i, j)$ where $i < j \leq h$ are observed. Therefore, with probability at least $1 - 2\delta/K$ (by union bound), the maximum number of times an arm $h$ is selected when the optimal arm's index is smaller than $h$ is $n_h^T$ such that

$$n_h^T = \left( \frac{C_1 \sqrt{d'} + C_2 \sqrt{\log(K^2/2\delta)}}{\lambda_\Sigma} \right)^2 + \frac{2}{\lambda_\Sigma} \left( \frac{2k_\mu \alpha_T}{\xi_h} \right)^2$$

$$= \left( \frac{C_1 \sqrt{d'} + C_2 \sqrt{\log(K^2/2\delta)}}{\lambda_\Sigma} \right)^2 + \frac{8}{\lambda_\Sigma} \left( \frac{k_\mu \alpha_T}{\xi_h} \right)^2$$

where $\xi_h = \min\limits_{i<h, t\geq 1} \xi_{ih}(x_t)$, $\lambda_\Sigma = \min\limits_{i<j\leq K} \lambda_{min}\left( \mathbb{E}\left[ \Phi_{ij}(X_s)\Phi_{ij}(X_s)^\top \right] \right)$ and $\alpha_T \geq \max\limits_{i<h} \alpha_{ih}^T$. By using union bound, we get following bound with probability at least $1 - \delta/2K$

$$\sum_{t=1}^{T} \mathbb{1}_{\{I_t = h, i_t^\star < h\}} \leq n_h^T. \tag{21}$$

From Eq. (21), using $\sum_{t=1}^{T} \mathbb{1}_{\{I_t = h, i_t^\star < h\}} \leq n_h^T$ and value of $n_h^T$, we get following upper bound on regret that holds with probability at least $1 - \delta$ by union bound

$$\mathfrak{R}_T \leq R_{max} \sum_{h=2}^{K} n_h^T = R_{max} \sum_{h=2}^{K} \left( \left( \frac{C_1 \sqrt{d'} + C_2 \sqrt{\log(K^2/2\delta)}}{\lambda_\Sigma} \right)^2 + \frac{8}{\lambda_\Sigma} \left( \frac{k_\mu \alpha_T}{\xi_h} \right)^2 \right).$$

Using $\alpha_T = \frac{2\sigma}{\kappa} \sqrt{\frac{d'}{2} \log\left(1 + 2T/d'\right) + \log\left(K^2/2\delta\right)}$ from Lemma 4 that ensures parameter $\theta_{ij}^\star$ bounds for all pairs $(i,j)$ holds with probability at least $1 - \delta/2K$ (by union bound) for $T > m$ where $m = C\lambda_\Sigma^{-2}\left(d + \log(k^2/2\delta)\right) + 2\lambda_\Sigma^{-1}$ such that $\lambda_{min}(V_{ij}^{m+1}) \geq 1$ for all pair $(i,j)$, we have

$$\mathfrak{R}_T \leq R_{max}\left( m + \sum_{h=2}^{K} n_h^T \right)$$

$$\implies \mathfrak{R}_T \leq R_{max}\left[ m + \sum_{h=2}^{K} \left( \left( \frac{C_1 \sqrt{d'} + C_2 \sqrt{\log\left(\frac{K^2}{2\delta}\right)}}{\lambda_\Sigma} \right)^2 \right. \right.$$

$$\left. \left. + \frac{16}{\lambda_\Sigma} \left( \frac{k_\mu \sigma}{\xi_h \kappa} \right)^2 \left( \frac{d'}{2} \log\left(1 + \frac{2T}{d'}\right) + \log\left(\frac{K^2}{2\delta}\right) \right) \right) \right]. \qquad \square$$

## B.6 Algorithm with Regularization Term

USS-PD uses forced exploration by selecting arm $K$ until the correlation matrix $V_{ij}^t$ is not invertible for all $(i,j)$ pairs of arms. Further, the minimum eigenvalue of $V_{ij}^t$ for all $(i,j)$ pairs is needed to be larger than $1$ so that bound given in Lemma 4 holds. Alternatively, $V_{ij}^t$ can be initialized by adding a regularization term (Abbasi-Yadkori et al., 2011; Zhang et al., 2016; Jun et al., 2017) to avoid forced exploration and then apply OFUL type analysis. We have given an algorithm named USS-PD-$\lambda$I which uses regularization term $\lambda I_{d'}$. However, its analysis still needed the minimum eigenvalue of the non-regularized part of the correlation matrix to become larger than some positive value (depends on $\lambda$ value), as shown in our next result.

---

**USS-PD-$\lambda$I Algorithm for Contextual USS using Pairwise Disagreement with $\lambda$I Initialization**

---

1: **Input:** Tuning parameters: $\delta \in (0,1)$ and $\lambda > 0$
2: Select arm $K$ for first context $x_1$
3: $\forall i < j \leq K$ : set $\overline{V}_{ij}^1 \leftarrow \lambda I_{d'} + \Phi_{ij}(x_1)\Phi_{ij}(x_1)^\top$ and update $\hat{\theta}_{ij}^1$ by solving Eq. (9)
4: **for** $t = 2, 3, \ldots$ **do**
5:     Receive context $x_t$. Set $i = 1$ and $I_t = 0$
6:     **do**
7:         Play arm $i$
8:         $\forall j \in [i+1, K]$ : compute $\tilde{p}_{ij}^{(t)} \leftarrow \mu\left( \Phi_{ij}(x_t)^\top \hat{\theta}_{ij}^{t-1} + \alpha_{ij}^{t-1} \|\Phi_{ij}(x_t)\|_{\left(\overline{V}_{ij}^{t-1}\right)^{-1}} \right)$
9:         If $\forall j \in [i+1, K] : C_j - C_i > \tilde{p}_{ij}^{(t)}$ or $i = K$ then set $I_t = i$ else set $i = i + 1$
10:     **while** $I_t = 0$
11:     Select arm $I_t$ and observe $Y_t^1, Y_t^2, \ldots, Y_t^{I_t}$
12:     $\forall i < j \leq I_t$ : update $\overline{V}_{ij}^t \leftarrow \overline{V}_{ij}^{t-1} + \Phi_{ij}(x_t)\Phi_{ij}(x_t)^\top$ and $\hat{\theta}_{ij}^t$ by solving Eq. (9)
13: **end for**

---

**Lemma 12.** *Let $\overline{V}_{ij}^t = \lambda \mathrm{I}_{d'} + V_{ij}^t$ for any $\lambda > 0$ and $\|\theta_{ij}\|_2 \leq S$ for all $(i,j)$ pair. Then for any $t > \min\{s : \forall i < j \ni \lambda_{\min}(V_{ij}(s)) \geq 2\lambda\}$, the following event holds for* USS-PD-$\lambda$I *with probability at least* $1 - 2\delta/K^2$,*

$$\left\|\hat{\theta}_{ij}^t - \theta_{ij}^\star\right\|_{\overline{V}_{ij}^t} \leq \beta_{ij}^t,$$

*where $\beta_{ij}^t = \frac{2\sigma}{\kappa}\sqrt{\frac{d'}{2}\log\left(1 + \frac{n_{ij}^t}{d'\lambda}\right) + \log\left(\frac{K^2}{2\delta}\right)} + 2\lambda^{1/2}S$.*

*Proof.* By using $\|Z\|_{A+B} \leq \|Z\|_A + \|Z\|_B$ , we have

$$\left\|\hat{\theta}_{ij}^t - \theta_{ij}^\star\right\|_{\overline{V}_{ij}^t} \leq \left\|\hat{\theta}_{ij}^t - \theta_{ij}^\star\right\|_{V_{ij}^t} + \left\|\hat{\theta}_{ij}^t - \theta_{ij}^\star\right\|_{\lambda \mathrm{I}_{d'}}. \quad (\text{as } \overline{V}_{ij}^t = \lambda \mathrm{I}_{d'} + V_{ij}^t)$$

When Eq. (9) is used for estimation of unknown parameter $\theta_{ij}^\star$ then by using Eq. (26) and Eq. (27) of Lemma 8 of Li et al. (2017), we have

$$\leq \frac{1}{\kappa}\left\|\sum_{s \in S_{ij}^t} \varepsilon_s \Phi_{ij}(x_s)\right\|_{(V_{ij}^t)^{-1}} + 2\lambda^{1/2}S. \quad (\text{as } \|\theta_{ij}\|_2 \leq S)$$

Sherman Morrison formula gives $\|Z\|_{(V_{ij}^t)^{-1}} \leq \left(1 - \frac{\lambda}{\lambda_{\min}(V_{ij}^t)}\right)^{-\frac{1}{2}} \|Z\|_{(\overline{V}_{ij}^t)^{-1}}$. Using it, we have

$$\left\|\hat{\theta}_{ij}^t - \theta_{ij}^\star\right\|_{\overline{V}_{ij}^t} \leq \frac{\left(1 - \frac{\lambda}{\lambda_{\min}(V_{ij}^t)}\right)^{-\frac{1}{2}}}{\kappa}\left\|\sum_{s \in S_{ij}^t} \varepsilon_s \Phi_{ij}(x_s)\right\|_{(\overline{V}_{ij}^t)^{-1}} + 2\lambda^{1/2}S.$$

Using upper bound on $\left\|\sum_{s \in S_{ij}^t} \varepsilon_s \Phi_{ij}(x_s)\right\|_{(\overline{V}_{ij}^t)^{-1}}$ as given in Theorem 1 of Abbasi-Yadkori et al. (2011), where $\varepsilon_s$ is $\sigma-$subGaussian random variable and holds with probability at least $1 - 2\delta/K^2$, we get

$$\left\|\hat{\theta}_{ij}^t - \theta_{ij}^\star\right\|_{\overline{V}_{ij}^t} \leq \frac{\left(1 - \frac{\lambda}{\lambda_{\min}(V_{ij}^t)}\right)^{-\frac{1}{2}}}{\kappa}\sqrt{2\sigma^2 \log\left(\frac{det(\overline{V}_{ij}^t)^{1/2}det(\lambda \mathrm{I}_{d'})^{-1/2}}{2\delta/K^2}\right)} + 2\lambda^{1/2}S$$

$$= \frac{\sigma\left(1 - \frac{\lambda}{\lambda_{\min}(V_{ij}^t)}\right)^{-\frac{1}{2}}}{\kappa}\sqrt{2\log\left(\frac{det(\overline{V}_{ij}^t)}{det(\lambda \mathrm{I}_{d'})}\right)^{\frac{1}{2}} + 2\log\left(\frac{K^2}{2\delta}\right)} + 2\lambda^{1/2}S.$$

By using Lemma 11 to upper bound $det(\overline{V}_{ij}^t)$, where $t > s$ with $a = 1$, and $n_{ij}^t \leq t$, we get

$$\left\|\hat{\theta}_{ij}^t - \theta_{ij}^\star\right\|_{\overline{V}_{ij}^t} \leq \frac{\sigma\left(1 - \frac{\lambda}{\lambda_{\min}(V_{ij}^t)}\right)^{-\frac{1}{2}}}{\kappa}\sqrt{d'\log\left(1 + \frac{n_{ij}^t}{d'\lambda}\right) + 2\log\left(\frac{K^2}{2\delta}\right)} + 2\lambda^{1/2}S. \quad (22)$$

As $t > s$ such that $\lambda_{\min}(V_{ij}(s)) \geq 2\lambda$, we have

$$\left\|\hat{\theta}_{ij}^t - \theta_{ij}^\star\right\|_{\overline{V}_{ij}^t} \leq \frac{2\sigma}{\kappa}\sqrt{\frac{d'}{2}\log\left(1 + \frac{n_{ij}^t}{d'\lambda}\right) + \log\left(\frac{K^2}{2\delta}\right)} + 2\lambda^{1/2}S = \beta_{ij}^t. \qquad \square$$

Note that if $\lambda_{\min}(V_{ij}(s)) < \lambda$ then $\left(1 - \frac{\lambda}{\lambda_{\min}(V_{ij}^t)}\right)^{-\frac{1}{2}}$ is not well defined and the bound given in Lemma 12 does not hold. Therefore, $\lambda_{\min}(V_{ij}(s))$ need to be at least greater than $\lambda$. Let $m' \doteq C\lambda_{\Sigma}^{-2}\left(d + \log(k^2/2\delta)\right) + 4\lambda_{\Sigma}^{-1}\lambda$ where $C > 0$ is the universal constant. Recall $R_{max} \doteq \max_{i \in [K], x \in \mathcal{X}}\left[C_i + \gamma_i(x) - (C_{i^\star} + \gamma_{i^\star}(x))\right]$, where $i^\star$ is the optimal arm for a context $x$. Now we state the regret bounds for USS-PD-$\lambda$I.

**Theorem 3.** *Let $\theta \in \Theta_{\mathrm{CWD}}$, $\lambda > 0$, $\delta \in (0,1)$, Assumption 1 holds, and $\xi_h = \min\limits_{t \geq 1} \xi_{i_t^\star h}(x_t)$. Then with probability at least $1 - 2\delta$, the regret of USS-PD-$\lambda$I for $T > m'$ contexts is upper bounded as*

$$\mathfrak{R}_T \leq R_{max}\left( m' + \sum_{h=2}^{K}\left( \left(\frac{C_1\sqrt{d'} + C_2\sqrt{\log\left(\frac{K^2}{2\delta}\right)}}{\lambda_\Sigma}\right)^2 + \frac{32\lambda}{\lambda_\Sigma}\left(\frac{k_\mu \sigma}{\xi_h \kappa}\right)^2 \right.\right.$$
$$\left.\left.\left(\sqrt{\frac{d'}{2}\log\left(1 + \frac{T}{d'\lambda}\right) + \log\left(\frac{K^2}{2\delta}\right)} + 2\lambda^{1/2}S\right)^2 \right)\right).$$

*Proof.* The proof follows similar steps as Theorem 2 by replacing $m$ by $m'$ and $\alpha_{ij}^T$ by $\beta_{ij}^T$. Using $\beta_{ij}^T = \sqrt{\frac{d'}{2}\log\left(1 + \frac{T}{d'\lambda}\right) + \log\left(\frac{K^2}{2\delta}\right)} + 2\lambda^{1/2}S$ completes the proof. $\qquad\square$

# C  Leftover details from Section 5

Since the parameter of each arm (classifier) is known to us (but not to the algorithm), the optimal arm $i_t^\star$ can be computed for every context. Therefore, we can also calculate the fraction of contexts for which WD property holds to a given cost vector. To verify WD property for a given context $x_t$, we first compute disagreement probability for each $(i,j)$ pair of classifiers as[3]

$$p_{ij}^{(t)} = \mu(x_t^\top \theta_i)(1 - \mu(x_t^\top \theta_j)) + \mu(x_t^\top \theta_j)(1 - \mu(x_t^\top \theta_i)).$$

When all $p_{ij}^{(t)}$ values and $i_t^\star$ are known, we can check whether a context $x_t$ satisfies WD property or not by using Eq. (3). For all problem instances derived from the synthetic dataset, the cost vector and the fraction of contexts for which WD property holds are given in Table 1.

| PI/Classifiers | Clf. 1 | Clf. 2 | Clf. 3 | Clf. 4 | Clf. 5 | WD fraction |
|---|---|---|---|---|---|---|
| Costs for PI 1 | 0.01 | 0.02 | 0.032 | 0.05 | 0.55 | 0.997 |
| Costs for PI 2 | 0.01 | 0.02 | 0.032 | 0.05 | 0.6 | 1.0 |
| Costs for PI 3 | 0.01 | 0.02 | 0.032 | 0.05 | 0.65 | 1.0 |
| Costs for PI 4 | 0.01 | 0.02 | 0.032 | 0.05 | 0.7 | 1.0 |

Table 1: Details of different problem instances (PIs) derived from synthetic datasets.

**Heart Disease dataset:** Each sample of the Heart Disease dataset has 12 features. We split the features into three subsets and train a logistic classifier on each subset. We associate 1st classifier with the first 7 features as input that include cholesterol readings, blood-sugar, and rest-ECG. The 2nd classifier, in addition to the 7 features, utilizes the thalach, exang, and oldpeak features; and the 3rd classifier uses all the features. For performance evaluation, the different values of costs are used in three problem instances for both real datasets, as given in Table 2. The PIMA diabetes dataset has 768 samples, whereas the Heart Disease dataset has only 297 samples. As 5000 contexts are used in our experiments, we select a sample in a round-robin fashion and give it as input to the algorithm.

| Values/ Classifiers | PIMA Indian Diabetes Dataset | | | | Heart Disease Dataset | | | |
|---|---|---|---|---|---|---|---|---|
| | Clf. 1 | Clf. 2 | Clf. 3 | WD Fraction | Clf. 1 | Clf. 2 | Clf. 3 | WD Fraction |
| Costs for PI 1 | 0.01 | 0.25 | 0.5 | 0.0692 | 0.01 | 0.25 | 0.5 | 0.1384 |
| Costs for PI 2 | 0.01 | 0.3 | 0.5 | 0.1192 | 0.01 | 0.3 | 0.5 | 0.1454 |
| Costs for PI 3 | 0.01 | 0.35 | 0.5 | 0.2204 | 0.01 | 0.35 | 0.5 | 0.2426 |

Table 2: Details of different problem instances (PIs) derived from real datasets.

**Experiments Results:** Through our experiments, we show that the stronger the CWD property (large value of $\xi$) for the problem instance, it is easier to identify the optimal arm and, hence, has lower regret, as shown in Fig. 2a. We also compare the performance of USS-PD with three baseline policies on problem instances derived from the Heart Disease dataset (same as the PIMA Indian Diabetes dataset). As expected, we observe that USS-PD outperforms the baseline policies, as shown in Fig. 2b. Note that we used $\delta = 0.05$ and $\sigma = 0.1$ in all experiments.

(a) Regret v/s CWD property ($\xi$).

(b) Total cost for PIMA Indian Diabetes dataset.

Figure 2: Performance of USS-PD.

## C.1 Realizable Setting

We consider the realizable case where all contexts satisfy Eq. (8) (by fixing $\theta_{ij}$ for each $(i, j)$ pair of arms) and WD property. Since WD holds, we can use Lemma 2 for finding the optimal arm. Note that the mean loss cannot be computed for this setting as we set parameters of disagreement probabilities instead of setting parameters for individual arms. We use an upper bound on the regret to evaluate the performance of USS-PD on the Synthetic dataset, as shown in Fig. 3. We repeat experiments 500 times to get a tighter confidence interval.

(a) Synthetic dataset with 4 classifiers where cost of using classifier $i$ in problem instance $j$ is $0.1 + (i - 1)(0.09 + (j - 1)0.01)$.

(b) Synthetic dataset with 5 classifiers where cost of using classifier $i$ for problem instance $j$ is $0.1 + (i - 1)(0.06 + (j - 1)0.01)$.

Figure 3: Performance of USS-PD for realizable setting where regret on y-axis is $\sum_{t=1}^{T} |C_{I_t} - C_{i_t^\star}| + p_{i_t^\star I_t}^{(t)}$, and it is an upper bound on the regret $\mathfrak{R}_T$ defined in Eq. (2). The value of $\xi$ largest for Case 1, and it decreases for subsequent cases.

**Regret used for Empirical Evaluation in Realizable Setting**
Since the error-rate of arms is unknown, the regret defined in Eq. (2) can not be computed. Hence we

define an alternative regret, which we call pseudo regret, as follows:

$$\mathfrak{R}_T^s = \sum_{t=1}^{T} \left[ C_{I_t} - C_{i_t^\star} + p_{i_t^\star I_t}^{(t)} \right].$$

It is easy to verify that the actual regret $\mathfrak{R}_T$ is upper bounded by above regret $\mathfrak{R}_T^s$ as shown follows:

$$\begin{aligned}
\mathfrak{R}_T &= \sum_{t=1}^{T} \left[ C_{I_t} + \gamma_{I_t}(x_t) - \left( C_{i_t^\star} + \gamma_{i_t^\star}(x_t) \right) \right] \\
&= \sum_{t=1}^{T} \left[ C_{I_t} - C_{i_t^\star} + \left( \gamma_{I_t}(x_t) - \gamma_{i_t^\star}(x_t) \right) \right] \\
&\leq \sum_{t=1}^{T} \left[ C_{I_t} - C_{i_t^\star} + p_{i_t^\star I_t}^{(t)} \right] \quad \text{(Using Lemma 1)} \\
\implies \mathfrak{R}_T &\leq \mathfrak{R}_T^s.
\end{aligned}$$

## C.2  Contextual Strong Dominance

We next introduce contextual strong dominance property of the problem instance.

**Definition 3** (Contextual Strong Dominance (CSD) property). *A problem instance is said to satisfy* CSD *property if for all contexts following is true:*

$$Y^i = Y \text{ for some } i \in [K] \implies Y^j = Y, \ \ \forall j \in [K] \setminus [i].$$

*We represent the set of all instances satisfies* CSD *property by* $\Theta_{\mathrm{CSD}}$.

The CSD property implies that if the feedback of an arm is the same as the true reward of a given context then, the feedback of all the arms in the subsequent stages of the cascade is also the same as the true reward of a given context.

When any problem instance satisfies CSD property, the value of $\mathbb{P}\left\{ Y_t^i = Y_t, Y_t^i \neq Y_t^j | X = x_t \right\} = 0$ for $j > i$. Therefore, for any $(i, j)$ pair of arms and context $x_t$ the following is true:

$$\forall j > i, \gamma_i(x_t) - \gamma_j(x_t) = \mathbb{P}\left\{ Y_t^i \neq Y_t^j | X = x_t \right\}.$$

The above equation implies that CWD property holds trivially for the problem instances that satisfy CSD property as the difference of mean losses is the same as the probability of disagreement between two arms(fix arm $i = i_t^\star$ for given context $x_t$).

## C.3  Effect of adding more arms on WD property

The performance of USS-PD can deteriorate as we increases as the number of arms. This is because the fraction of contexts that satisfy WD property can decrease with the increase in the number of arms. To see that, consider a contextual USS problem instance with three arms where arm 1 has cost 0.1, arm 2 has cost 0.2, and arm 3 has cost 0.3. Let there be two contexts $x_1$ and $x_2$ such that classifier 2 is an optimal classifier for context $x_1$ and classifier 3 for the context $x_2$, and both contexts satisfy WD property. When a new arm is added at the end of the classifiers cascade without changing the optimal arm for the contexts, let $p_{24}^{(1)}$ be the disagreement probability for classifier 2 and 4 for context $x_1$ and $p_{34}^{(2)}$ be the disagreement probability for classifier 3 and 4 for context $x_2$. It is easy to verify that if cost of using classifier 4 is less than $\min\{0.2 + p_{24}^{(1)}, 0.3 + p_{34}^{(2)}\}$ then both contexts will not satisfy WD property.