[Reviews · NeurIPS 2020]

Review 1

Summary and Contributions: In this paper the authors propose an extension of the Unsupervised Sequential Selection problem to contextual input. They introduce the notion of Contextual Weak Dominance, and show that it is sufficient to solve the problem. They propose an algorithm, USS-PD, and show that it is able to achieve logarithmic regret under the additional hypothesis xi >0. Finally, they evaluate the performance of USS-PD on both real and synthetic datasets.

Strengths: Overall I think that this paper presents some significant contributions to USS : the authors prove that their condition, the contextual weak dominance, is sufficient to identify the best arm, and that in this case it is sufficient to estimate the disagreement between arms to achieve this objective. Their analysis of the regret of USS-PD is interesting and the empirical evaluation is convincing.

Weaknesses: - The formulation of some results are mildly confusing : for instance the result of Lemma 1 should be equality, and does not require j > i, as shown by the authors in their proof. It is unclear to me why the authors chose to only report a weaker result. - Technically it is possible to satisfy CWD and have \inf ( \xi) = 0. I think the paper could benefit from some insights of the authors regarding this particular case. - While this setting, Contextual USS, is clearly new (to my knowledge), most of the tools and results presented in this article are derived from existing results in USS and GLM. However, their extension to the contextual USS setting appears to be non trivial to me, so this work contribution is significant.

Correctness: All claims and theoretical results presented are proven, and the empirical methodology is correct.

Clarity: The paper is well written, and the proofs are detailed enough to be easy to follow.

Relation to Prior Work: The novelty of this work is clearly discussed in the paper.

Reproducibility: Yes

Additional Feedback:


Review 2

Summary and Contributions: [Please find the update in the "Additional feedback section"] The authors study a problem of selecting a classifier for making predictions from the collection of classifiers with various cost and reliability, called "Contextual Unsupervised Sequential Selection". This work elaborates on the recent study of Unsupervised Sequential Selection (without taking into account contexts). To model context-conditional correlations between arms the authors adopt the idea of GLM bandits.

Strengths: The paper is well-organized and the problem is clearly stated. The work responds to the call from the prior art regarding studying contextual version of the USS problem, so it covers a blank space in this research area.

Weaknesses: The idea might look incremental: what are the main challenges solved by this work apart the combination of USS and GLM bandits? The main weakness is experimental section that lacks comparison with baselines and other SOTA methods.

Correctness: The methodology is correct in general, but I have not checked math. details

Clarity: The paper is well written.

Relation to Prior Work: The prior art is clearly discussed.

Reproducibility: No

Additional Feedback: Update after rebuttal: Based from the authors feedback and other reviews I can see the paper is not that straightforward from the mathematical viewpoint, but I am not a deep specialist in bandits theory to acknowledge these contributions. However, I still think the paper lacks experiments. "To the best of our knowledge, we are first to consider the contextual USS problem, so there is no state-of-the-art (SOTA)" > There are methods for non-contextual USS problem, but none of them was considered as a baseline. Only weak baselines were provided. Therefore I will slightly increase the overall score of this submission.


Review 3

Summary and Contributions: This paper studies Contextual Unsupervised Sequential Selection (USS) problem, which is a new variant of the stochastic contextual problem. Compared with vanilla-USS (Verma et al., 2019), the decision maker can observe an additional context during each round, which is a new setting not studied before. The goal of the problem is to learn a decision rule such that the total expected loss is minimized. A new algorithm with sub-linear regret under the assumpsion that the problem instance satisfies `Contextual Weak Dominance` is proposed.

Strengths: An algorithm suffers O(\log T) regret is proposed and it is validated by experiments on both synthetic and real datasets.

Weaknesses: No lower bound is presented.

Correctness: I am not an expert in this area. From the presentation, the claims look reasonable to me.

Clarity: The paper is well written and the theorems are clearly stated.

Relation to Prior Work: This paper clearly states the difference between previous contributions.

Reproducibility: Yes

Additional Feedback: Overall the paper is well written with good presentation. People in multi-armed bandits fieid may be interested in this problem. One drawback of the paper is that no lower bound is provided and it is hard to tell whether the variables hidden from big-O in corollary 1 can be optimized. In equation (2), \lambda_{I_t} is missing and it is explained in line 126 why \lambda_{I_t} is missing. It is better if it can be put before equation (2). Typos: Line 85: generate -> generates Line 89: denote -> denotes ==== After rebuttal ==== The author feedback and the reviews of the other reviewers did not change my positive score about the paper.


Review 4

Summary and Contributions: In this paper, the contextual unsupervised sequential selection problem is studied. Different from the classical contextual bandit algorithm, the learner cannot observe the losses when the arms are pulled. Instead, an ordered relationship is assumed among the arms which are named the contextual weak dominance property. Relying on this assumption, the true loss can be equivalently reduced to the disagreement among arms. A context-based learning algorithm is proposed under this condition, whose theoretical guarantee is proved and empirical performance is verified through experiments.

Strengths: - The paper studies a new setting for contextual bandit learning, which is a topic with theoretical and practical importance. - The theoretical results are grounded with rigorous analysis. The proposed approach is verified through proper experimental results.

Weaknesses: - The contribution seems to be incremental comparing to the previous works (see detailed discussions below)

Correctness: As far as I can see, the claims and methods are correct both in theoretical and algorithmic perspectives.

Clarity: The paper is overall clearly written. The contributions are clearly described, and the analysis and algorithms are detailed introduced.

Relation to Prior Work: As far as I can see, the literature survey is thorough, and the relationships to previous studies are clearly explained.

Reproducibility: Yes

Additional Feedback: - The main concern for me is the technical novelty comparing to previous studies, especially [1]. In [1], the exactly same weak dominance property is also introduced, and the Theorem 1 in [1] is the same to Theorem 1 in the paper. By this theorem, extending classical GLM algorithm to the current setting seems to be somehow straightforward. I think it is useful to discuss the main technical challenges the proposed approach overcomes in the paper. - To make the setting more novel to previous study, it may be interesting to refine the assumptions on models to consider new constraints. For example, to consider the setting in which the context received in each round contain private information which needs to be protected. - I wonder if the feedback y can be real values instead of only binary values. This can make the setting more general. [1] Online Algorithm for Unsupervised Sensor Selection, AISTATS’19. --- after rebuttal My concern lies in the technical significance of extending the USS problem to the GLB setting. After reading all the reviews and the author feedback, I still find the assumption similar to the previous work as pointed out in my review. Even though I do think the paper studies an important real problem, I keep my score unchanged.

[Author Response · NeurIPS 2020]

We thank reviewers for detailed comments and suggestions. We will address all comments in the revision. In this work,
we consider a novel unsupervised stochastic contextual bandits problem. In follow-on work, we will study relevant
open questions like, lower bounds, private information, and real-valued feedback, pointed out by reviewers.

**Novelty.** The novelty of our work, as such, is a combination of novel modeling principles to account for unsupervised
contextual sequential selection, as well as subsequent method and analysis, and experimentation. The earlier work
(Verma et al. AIStats'19) considered the problem of learning an optimal action but ignored the contextual information.
In this work, we incorporated the contextual information, which is readily available in many applications. Exploiting
the *real-valued* contextual information (features) for improving the arm selection strategy is non-trivial due to the
unsupervised nature of the problem where the standard analysis of contextual bandits does not apply. We made necessary
modeling assumptions leveraging GLM models and extended the existing definitions to address the learnability issues
in the new setup. However, the problem still requires new ideas and analysis methods to derive an efficient algorithm
and poses new technical challenges for analysis.

**Response to common comments of Reviewers $2$ and $4$:**
`The idea might look incremental.  What are the main challenges solved by this work?  ` :
We respectfully disagree. The new ideas and technical challenges addressed in this work are as follows:

**New ideas:** USS-UCB (Verma et al. AIStats'19) uses two-sided test derived from Eq. (6) and Eq. (7) to identify an
optimal arm. In contrast, our work identifies that a one-sided test is enough to learn an optimal arm with contextual
information. We exploit this idea to come up with simpler algorithms. The difference in tests used in earlier and our
work becomes apparent by comparing arm selection strategy in USS-PD (line 9) with that of USS-UCB (lines 7-9)
where two sets $\hat{B}_t^h$ and $\hat{B}_t^l$ and their interaction needs to be computed. However, this simplification in USS-UCB throws
some challenges in the analysis that do not arise with two-sided test, but we carefully handle it (see lines 488-492).

**Technical Challenges**
1. GLM bandits are well studied but require reward or loss information. In the USS setup, loss of selected arm can
not be observed; hence finding the optimal arm is challenging. We have shown that if problem instance satisfies
contextual weak dominance (CWD) property, then the pairwise disagreement between arms can be used to estimate
context-dependent disagreement probability, and that can be used to find an optimal arm for a given context.
2. Regret analysis of GLM bandits hinges on bounding the instantaneous regret in each round, which is tied to the
estimation error of the GLM parameters. Due to the unsupervised setting and cascade structure, this way of
bounding regret does not work in our setup. Our analysis goes by bounding the number of pulls of the sub-optimal
arms. However, unlike standard bandits, we have to distinguish whether the sub-optimal arm pulled by USS-PD is
on the 'left' or 'right' of the optimal arm in the cascade. It requires our analysis to carefully handle both the cases
(see Lemma 5 and 6). Since USS-PD uses a similar MLE estimator for parameter estimation as in GLM bandits,
we only adapt their asymptotic normality results, the other steps of bounds are new in our work.
3. Though it is not reported in our work, we did try several other models and analysis approaches to solve the USS
problem with contextual information. However, due to the weak feedback structure of the problem, the other
methods are not amenable for analysis. Our final presentation is a model and analysis that is clean and complete.
For example, in the appendix, we point out that analysis based on Optimism in the Face of Uncertainty for Linear
bandits (OFUL) method with a regularizer did not go through without making more assumptions.

**Response to Reviewer $1$:**
`Lemma 1 should be equality, and does not require` $j > i$: Thanks for catching the typo. We will fix it.

**Response to Reviewer $2$:**
`Experiments lacks comparison with baselines and other SOTA methods:`
To the best of our knowledge, we are first to consider the contextual USS problem, so there is no state-of-the-art (SOTA)
method. In our experiments, we have considered baseline policies that select either a fixed arm or a uniformly random
arm in each round (see Figure 1c). We also compare USS-PD's performance with the policy that can observe the true
loss (see Figure 1b).

**Response to Reviewer $3$:**
`In equation (2),` $\lambda_{I_t}$ `is missing`: Thank you for catching the mistake. We will correct it.

**Response to Reviewer $4$:**
`In [1], the exactly same weak dominance property is introduced and Theorem 1 is same:`
We **disagree** on both. In this paper, the WD property is context-dependent, whereas the contextual information is
ignored in [1]. This new definition has nuances. First, Eq. 4 points to the fact that examples can be partitioned
based on strength of CWD, a situation that does not arise in [1]. Additionally, we allow for instances to violate CWD
property, and present algorithms that are agnostic to the presence of these instances. As for Theorem 1, although it
bears similarities with the previous work but it requires adaptation to the contextual setting.

[Meta-Review · NeurIPS 2020]

While the reviewers find the paper interesting and the results correct, there were some concerns regarding the novelty of the techniques beyond Verma et al AISTATS 2019 and GLM bandit literature. I am only partially convinced by the author's claim of novelty in their response. While the contextual setting is novel in unsupervised sequential selection, the techniques used for this extension under GLM and CWD assumptions seem incremental. However, the problem studied seems important and the results are new. Therefore, I am open to accepting this paper.